# $Dbh^+$ catecholaminergic cardiomyocytes contribute to the structure and function of the cardiac conduction system in murine heart

Tianyi Sun[1,13], Alexander Grassam-Rowe[1,13], Zhaoli Pu[2,3,13], Yangpeng Li[2,3], Huiying Ren[2,4], Yanru An[5], Xinyu Guo[6], Wei Hu[7], Ying Liu[8], Yuqing Zheng[2,3], Zhu Liu[2,3], Kun Kou[2,3], Xianhong Ou[2], Tangting Chen[2], Xuehui Fan[2], Yangyang Liu[8], Shu Tu[1], Yu He[1], Yue Ren[1], Ao Chen[5], Zhouchun Shang[5], Zhidao Xia[9], Lucile Miquerol[10], Nicola Smart[11], Henggui Zhang[7,12], Xiaoqiu Tan[2,3,4] ✉, Weinian Shou[8] ✉ & Ming Lei[1,2] ✉

The heterogeneity of functional cardiomyocytes arises during heart development, which is essential to the complex and highly coordinated cardiac physiological function. Yet the biological and physiological identities and the origin of the specialized cardiomyocyte populations have not been fully comprehended. Here we report a previously unrecognised population of cardiomyocytes expressing *Dbh* gene encoding dopamine beta-hydroxylase in murine heart. We determined how these myocytes are distributed across the heart by utilising advanced single-cell and spatial transcriptomic analyses, genetic fate mapping and molecular imaging with computational reconstruction. We demonstrated that they form the key functional components of the cardiac conduction system by using optogenetic electrophysiology and conditional cardiomyocyte *Dbh* gene deletion models. We revealed their close relationship with sympathetic innervation during cardiac conduction system formation. Our study thus provides new insights into the development and heterogeneity of the mammalian cardiac conduction system by revealing a new cardiomyocyte population with potential catecholaminergic endocrine function.

The cardiac conduction system (CCS) plays an essential role in cardiac physiological function by initiating and coordinating cardiac excitation and contraction. The CCS comprises a set of distinct and specialized components, including the pacemaker sinoatrial node (SAN), the coordinating atrioventricular node (AVN) and the rapid transmission ventricular His–Purkinje system. Failure in correct development of the CCS components leads to several fast (e.g., Wolff-Parkinson-White syndrome) and slow dysrhythmic (sick sinus syndrome and atrioventricular block) diseases[1]. Gaining insight into the

morphogenesis and maturation, and the physiological nature of cell-type heterogeneities of the CCS during the developmental process is critical for understanding the pathogenesis of these dysrhythmic diseases[1]. However, our understanding of when and how diverse cell types arise to form different components of the CCS remains limited[2].

In murine heart, pacemaker activity can be detected as early as embryonic day (E8.0)[3]. However, the formation of the CCS is not completed until perinatal stages[2,4]. The origin of the cell, cellular differentiation, and the maturation of the various components of the

---

CCS, in particular the ventricular CCS, are incompletely understood. In murine hearts, it has been accepted that the Purkinje fibre (PKJ) network arises from developing trabecular cardiomyocytes through endocardially-derived inductive signals[5–8]. Recently, it has also been suggested that a polyclonal PKJ network forms by progressive recruitment of conductive precursors to this scaffold from a pool of bipotent progenitors[9].

On the other hand, catecholamines, the canonical sympathetic neurotransmitter, are known to be essential for cardiac development: mice deficient in catecholamine synthetic enzymes tyrosine hydroxylase (*Th*)[10] and dopamine beta-hydroxylase *(Dbh)*[11,12], died in utero with cardiac developmental defects. There is also mixed evidence around the role of catecholamines in establishing and maintaining normal cardiac rhythm[13–16]. However, the true importance and source of these catecholamines for the CCS development remain uncertain[16].

Recent refinement of transcriptomic technologies including single-cell RNA sequencing (scRNA-Seq) and spatial enhanced resolution omics sequencing (Stereo-seq)[17,18] has facilitated the interrogation of cell populations with increasing classification power, revealing and characterizing novel cell types. Several studies have applied such approaches, particularly scRNA-Seq, to investigate cardiogenesis, focusing on subpopulations of cells isolated using predefined genes[19–22]. Given the complexity and incomplete understanding of the CCS morphogenesis and maturation, it is important to establish a molecular approach that enables simultaneous analysis of global organ-wide spatial gene expression patterns without biasing against cellular heterogeneity. We thus co-opted Stereo-seq, a newly established high-resolution large-field spatially resolved transcriptomics technique[17,18], with scRNA-Seq, to map the transcriptional landscape of the CCS formation and maturation in the murine heart. This led to the identification of an unreported cardiomyocyte population expressing *Dbh*. Our subsequent lineage tracing analysis revealed that *Dbh*+-derived CMs first emerged in the sinus venosus at E12.5, subsequently contributed to both atrial and ventricular CCS components in the process of cardiogenesis and became more prominent in the CCS from perinatal stages to adults. Using optogenetic electrophysiological interrogation of *Dbh^Cre*/ChR2-tdTomato mice, we further confirmed that *Dbh*+-derived CMs function as part of the ventricular CCS. We then performed electrophysiological characterization of cardiac conduction using ex vivo isolated hearts from a *Dbh^cko(α-MHC^Cre/Dbh-flox)* cardiomyocyte-specific *Dbh* deletion model to confirm that *Dbh^cko* hearts have slowed atrioventricular conduction. Furthermore, catecholaminergic-type vesicles were identified in adult cardiomyocytes in the *Dbh*+-CMs-rich atrioventricular junction. Collectively, our data provide a transcriptomic landscape of the CCS in the murine heart which reveals *Dbh*-expressing cardiomyocytes contributing to the formation and function of the CCS in the mammalian heart.

## Results

### Integrated transcriptomic analyses identified a population of *Dbh*+ cardiomyocytes in the murine heart

We first sought to characterize the cellular composition of the developing murine heart from middle-to-late embryonic (E8.5, E10.5, E12.5, E14.5, E16.5) and postnatal (P3) stages, by combining scRNA-Seq and Stereo-seq (Fig. 1a). Our workflow combined current best practices with recent computational advances (Fig. 1a and Fig. S1).

We established the major cell populations across the developing murine heart. With the use of scRNA-Seq, by 10×Genomics Illumina HiSeq PE150 (Fig. 1a), following pre-processing and library quality control, our data had mean UMI and gene counts of 6417 and 1482 respectively across 175237 cells in our scRNA-Seq (Fig. S2). We identified a range of cell types across the developing hearts and embryos (Fig. S3). Initial analysis suggested that the scRNA-Seq identified 4 overarching populations – early embryonic tissue (e.g., endodermal *Afp*^hi, mesodermal *Prrx1*^hi, or neural *Sox2*^hi), mixed non-myocyte tissue

(*Col1a1*^hi, *Fbln2*^hi, *Emcn*^hi, *Upk3b*^hi, *Lmod1*^hi), and ventricular (*Myl2*^hi, *Myl7*^lo, *Myh7*^hi, *Kcne1*^hi) and atrial (*Myl7*^hi, *Myh6*^hi, *Nppa*^hi, *Sln*^hi) cardiomyocyte populations across 26 cell types (Fig. S3a, Fig. S4, Fig. S5a–d, Fig. S6a, Data S1). Thus, our scRNA-Seq could sample the expected cellular composition of the developing heart at a gross level.

We could discriminate major populations and subpopulations of working and non-working cardiomyocytes across our scRNA-Seq dataset of the developing murine heart. To better understand cardiomyocyte populations in the developing heart, we undertook more extensive analysis of the cardiomyocyte lineage. To focus on identifying heterogeneity within and between differentiated and transitory cell states in the cardiomyocyte-lineage, we selected cell types mapping the expected developmental lineage of cardiomyocytes, from mesodermal cardiac progenitors through to neonatal cardiomyocytes to best discriminate transcriptional similarities and differences. We utilized a recently developed local and global structure-preserving dimensionality reduction technique (PHATE)[23] and "spectral-like" unsupervised clustering to identify 15 cardiomyocyte lineage cell types over $n = 107,705$ cells from earlier cardiac progenitors across to more mature cardiomyocyte cell types over the 3 dimensional PHATE space (Data S2), which can be taken as an approximation of pseudotime cell-state progression, given the manifold structure-preserving nature of the PHATE algorithm[23]. We observed the "development" from the mesodermal cardiogenic heart fields (*Hand1*^hi, *Wnt2*^hi, *Osr1*^hi), through primary heart tube (*Acta2*^hi, *Tagln*^hi, *Pmp22*^hi) formation, to immature ventricular (*Myl2*+, *Myl7*^lo, *Actc1*+, *Tnni1*^hi, *Tnni3*^lo) and atrial (*Actc1*+, *Sln*+, *Myl7*^hi) cardiomyocytes, and then through into more mature atrial (*Myl7*^hi, *Sln*^hi, *Nppa*^hi, *Tnni3*^hi) and ventricular (*Myh7*^hi, *Myh6*^lo, *Myl2*^hi, *Kcne1*^hi, *Tnni3*^hi) cardiomyocytes (Fig. 1b, Fig. 1d, Fig. S6b, Fig. S7, Fig. S8a–c, Data S2, Table S1). Interestingly, we saw that much of the CCS formed clusters distinct in 3D PHATE space from their respective most similar working cardiomyocyte populations (Fig. 1b, Data S2). For example, Purkinje Fibres (PKJs) compared with the trabecular ventricular myocardium. With use of the "spectral-like" clustering algorithm[23], and cluster labelling based on gene expression profiles (Fig. 1d, Fig. S6b, Fig. S9, Data S2, Table S1) we could resolve the CCS from our mixed scRNA-Seq, whereas many prior scRNA-Seq studies[19–22] did not discriminate separate CCS populations apart from those using CCS-biased sampling techniques[24,25]. Therefore, we used novel analytical approaches to better understand and reveal heterogeneity within the developing mouse cardiomyocyte lineage.

We identified previously unreported dopamine beta-hydroxylase (*Dbh*) expressing cardiomyocytes (*Dbh*+-CMs) populations from within the cardiomyocyte lineage (Fig. 1c). We observed *Dbh* expression across both working and non-working (i.e. pacemaker/conducting) cardiomyocyte populations (Fig. 1c). The major *Dbh*+-CM cell types were: atrial and ventricular cardiomyocytes; early and more mature ventricular trabecular cardiomyocytes; sinoatrial and atrioventricular nodal cardiomyocytes; and non-specific atrial cardiac conduction system and Purkinje fibres, although we did identify a small number of cells that belonged to earlier developing cardiomyocyte populations (Fig. S10, Fig. S11, Data S3). Overall, there were $n = 2591$ *Dbh*+-CM, at 2.43% of all cardiomyocyte lineage cells from E8.5, through to P3. However, significant numbers of *Dbh* + cells only became apparent beyond E10.5 (Data S3). We confirmed expression of expected specific pan-cardiomyocyte markers such as *Tnnt2* (Fig. S6ci), and CCS-specific markers (Fig. S6cii), within the *Dbh*+ population. Strikingly, although *Dbh*+-CM represented only small percentages of the total working cardiomyocyte populations (AM: 2.13%, VM: 3.90%, VM-trab: 4.18%, and eVM-trab: 2.26%) (Fig. S11), the CCS appeared to be over-represented in *Dbh*+ cardiomyocytes as percentages of their respective total cardiomyocyte lineage numbers (PKJ: 12.76%, AVN: 12.14%, SAN: 8.39%, and AM-CCS: 3.82%) (Fig. S11). Pearson's Chi-Squared test for independence confirms that cell type, within cardiomyocytes, and *Dbh* expression (positive or negative) are not independent ($\chi^2 = 1155.2$,

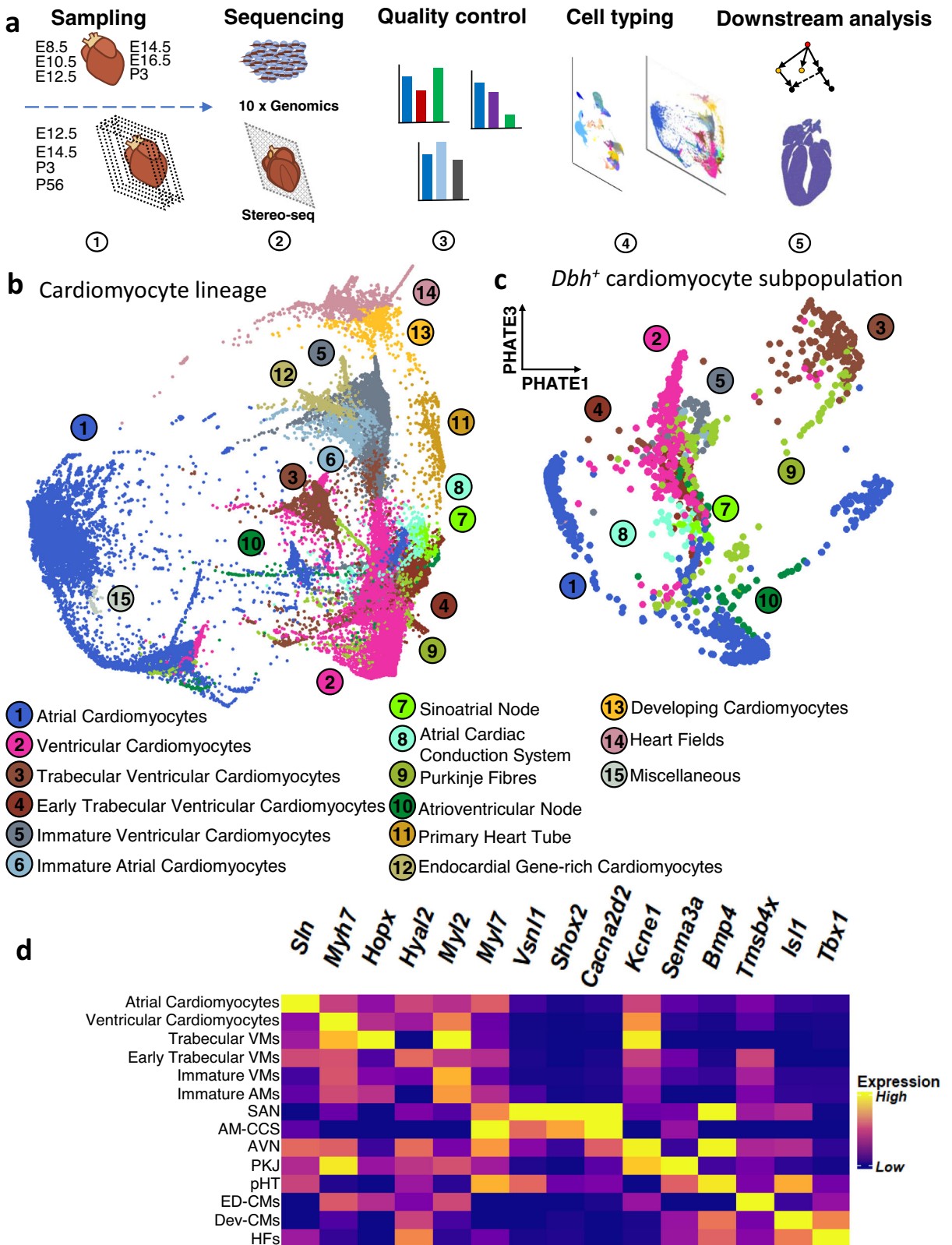

**a** Sampling | Sequencing | Quality control | Cell typing | Downstream analysis

E8.5 E14.5
E10.5 E16.5
E12.5 P3

E12.5
E14.5
P3
P56

10 x Genomics

Stereo-seq

① ② ③ ④ ⑤

**b** Cardiomyocyte lineage

**c** *Dbh⁺* cardiomyocyte subpopulation

PHATE3
PHATE1

1. Atrial Cardiomyocytes
2. Ventricular Cardiomyocytes
3. Trabecular Ventricular Cardiomyocytes
4. Early Trabecular Ventricular Cardiomyocytes
5. Immature Ventricular Cardiomyocytes
6. Immature Atrial Cardiomyocytes
7. Sinoatrial Node
8. Atrial Cardiac Conduction System
9. Purkinje Fibres
10. Atrioventricular Node
11. Primary Heart Tube
12. Endocardial Gene-rich Cardiomyocytes
13. Developing Cardiomyocytes
14. Heart Fields
15. Miscellaneous

**d**

Columns: Sln, Myh7, Hopx, Hyal2, Myl2, Myl7, Vsnl1, Shox2, Cacna2d2, Kcne1, Sema3a, Bmp4, Tmsb4x, Isl1, Tbx1

Rows: Atrial Cardiomyocytes, Ventricular Cardiomyocytes, Trabecular VMs, Early Trabecular VMs, Immature VMs, Immature AMs, SAN, AM-CCS, AVN, PKJ, pHT, ED-CMs, Dev-CMs, HFs

Expression
High
Low

df = 9, *p* = 5.8e−243, *n* = 103008). To summarize, we identified previously unreported transcriptomic expression of a catecholamine biosynthetic enzyme within developing murine cardiomyocytes, and its association with the CCS.

To facilitate future hypothesis generation, we also provide a publicly available web resource to visually explore our transcriptional landscapes in a cell-wise manner (Data S1−3), allowing examination of global and local cell distributions, as well as identifying a range of quality control metrics, in the developing murine heart.

**Stereo-seq recapitulated our identification of *Dbh*-expressing cardiomyocytes across the developing and mature murine heart**
To validate the identification of *Dbh⁺*-CMs, we used a new nanoscale resolution spatial transcriptomics technology Stereo-seq[17,18], to

**Fig. 1 | Single-cell RNA sequencing identified a *Dbh*+ subpopulation within the cellular landscape of the murine cardiomyocyte lineage. a** Summary of workflow: whole embryos (E8.5, E10.5) or whole hearts (E12.5, E14.5, E16.5, P3) were sampled for single cell RNA sequencing. Slices of whole heart from E12.5, E14.5, and P3 were sampled for Stereo-seq spatial transcriptomics. We used 10X Genomics and Stereo-seq for sequencing. We performed quality control before we clustered and labelled cells based on their gene expression. We performed a range of downstream analyses with both single-cell RNA sequencing and Stereo-seq data. **b** PHATE plot of the cellular landscape of the cardiomyocyte lineage. Numbered labels indicate the cell type with the corresponding colour on the PHATE plot, with full names below. (See Supplementary Data 2 for interactive version). **c** PHATE plot of the *Dbh*+ cardiomyocyte subpopulation in the developing murine heart. Numbered labels indicate the cell type with the corresponding colour on the PHATE plot, with full names below and are common with panel b. (See Supplementary Data 3 for interactive version). **d** Heatmap demonstrating selected marker genes for each population identified in the cardiomyocyte lineage. Yellow indicates high expression and purple indicates low expression. Gene expression values were normalised across heatmap rows, and then across columns. VMs ventricular cardiomyocytes, AMs atrial cardiomyocytes, SAN sinoatrial node, AMCCS atrial cardiac conduction system, AVN atrioventricular node, PKJ Purkinje fibres, pHT primary heart tube, ED-CMs endocardial gene-rich cardiomyocytes, Dev-CMs developing cardiomyocytes, HFs heart fields.

analyze this *Dbh*+ cardiomyocyte population, including its spatial distribution, in the developing and mature murine hearts. We established a *Dbh*+ cell lineage tracing reporter line, *Dbh*Cre/Rosa26-tdTomato mice, for genetic fate mapping of *Dbh*+-CMs (Fig. 2a). Following sampling, library preparation, sequencing and quality control, we had at least *n* = 3 sections for hearts from *Dbh*Cre/Rosa26-tdTomato mice at E12.5, E14.5, P3, and P56, respectively. DNA-labelled nanoballs were binned at 20 × 20 (~14 μm spot diameter) for E12.5, E14.5 and P3, and at 50 × 50 (~35 μm spot diameter) for P56 (Fig. S12a).

Stereo-seq provided a spatiotemporal transcriptomic map of the murine heart with a panoramic field of view and in situ cellular resolution of the CCS. First, we sought to confirm Stereo-seq could faithfully recapitulate known biology in the specific context of the developing murine heart. We were able to resolve the expected changes in gene expression and spatial distribution across both the developing and adult mouse heart (Fig. S13a–c). Confident in examining global changes in gene expression across our Stereo-seq dataset, we sought to ensure this matched with the appropriate cell type co-expression of genes. We identified a range of working and non-working cardiomyocyte populations in our analysis of stages E12.5, E14.5, and P3, alongside expected non-cardiomyocyte populations (Fig. 2b, c, Fig. S14). We omitted P56 from cell type classification to ensure comparability with our scRNA-seq data, which does not extend beyond P3. To minimize the potential of bias in our cell type classification; we also used additional supervised learning workflows to classify cells using our scRNA-Seq dataset as a reference for cell-type specific gene expression (Table S2)[26,27]. Overall, this supervised classification largely agreed with our cell types identified (Fig. S15). However, we note that "AM-CCS", mapped in a non-specific manner across the atrial myocardium (Fig. S15), likely reflecting the weaker expression of CCS markers in AM-CCS, compared to the SAN/AVN, but with expression of more generic atrial markers (Fig. S9). We believe the AM-CCS likely represents either some transitional sinus node population, or indeed inter-atrial conduction pathways (Fig. S9). Those cell types with imperfect integration often appear to have mapped to cell types with expectedly similar transcriptomic signals; analysis of the robust cell type decomposition (RCTD) scores for certain populations suggests that these cell types may still be identified by this supervised learning technique, but that the final highest predictive score for each cell might be obscuring the complexity of similar or even overlapping cell types being captured in each bin (Fig. S15). In sum, Stereo-seq could resolve cell types across the developing heart, down to a quasi-single cell resolution, within the context of a whole-organ sample.

Stereo-seq supported our identification of *Dbh* expression in cardiomyocytes across the developing heart. Following our cell type identification, we proceeded to investigate *Dbh* real-time expression and lineage tracing. We could readily observe *Dbh* expression across the myocardium at all stages observed (Fig. 2ci-ii, Fig. S13d). We recapitulated our earlier findings in our scRNA-Seq data that *Dbh* expression was across both working and non-working cardiomyocyte cell types from E12.5 through to P3 (Fig. 2ci). However, *Dbh* expression appeared particularly high in the His/Bundle Branch part of the proximal ventricular CCS (Fig. 2ci). Concordantly, we saw *Dbh* spatial

expression overlapping with a number of ventricular CCS markers, such as *Cntn2*, *Cacna2d2*, *Gja5*, *Hcn4*, and *Id2*, as shown in a representative slice from P3 in Fig. 2cii. The spatial co-localisation of *Dbh* and CCS marker genes identified by Stereo-seq provides additional support to the association of *Dbh* with the developing CCS, particularly in the ventricular CCS.

Stereo-seq enabled spatiotemporal genetic fate mapping of *Dbh*+-derived cells across the murine heart. We also sought to understand the distribution of the *Dbh* lineage beyond real-time expression by using our *Dbh*Cre/Rosa26-tdTomato reporter line. Unfortunately, tdTomato-coding sequences in our generated libraries were not readily detectable due to the use of 3′-end 100 base pair sequencing, thus the expression levels of *Wpre* (expressed at 3′-end of the tdTomato reporter cassette construct) (Fig. 2a) were used as a proxy for tdTomato expression in our sequencing data, which represent *Dbh*+-derived and potential *Dbh*+ cells. Overall, we identified the *Dbh*+-derived cardiomyocytes (*Dbh*+-derived CMs) lineage was associated with the CCS in a somewhat similar manner as real time *Dbh* expression. *Wpre* expression was highest in the CCS (Fig. 2ci). However, we did also observe *Wpre* expression across the non-working and working myocardium from E12.5 through to P56 (Fig. 2cii, Fig. S13d). Of note, *Wpre* expression signal appeared stronger in the SAN than in the ventricular CCS, which was the opposite of *Dbh* expression observed in our Stereo-seq data and scRNA-Seq data (Fig. 2ci, Fig. S11). This likely reflects differences in the temporal expression dynamics between endogenous *Dbh* and the reporter construct, such that cells in the atrial CCS (e.g. SAN), might have ceased to express *Dbh* but continue to express *Wpre* as descendants of once *Dbh*+ progenitors, or in their own changing transcriptional programs. The developmental multi-stage and spatially-resolved tracking of our tdTomato lineage reporter by Stereo-seq supported the expression of *Dbh* across cardiomyocyte lineages, but with particular association with the CCS.

Cardiomyocyte *Dbh* expression persisted into the adult murine heart. To confirm whether these developmental *Dbh*+- and/or *Wpre*+-cardiomyocyte populations were still present in adulthood, we examined adult (P56) heart slices prepared from the *Dbh*Cre/Rosa26-tdTomato mice through Stereo-seq. At P56, *Wpre*+ cardiomyocytes were numerous in the right atrium, SAN, AVN, and His-Purkinje network similar to in development (Fig. S13d). A number of *Dbh*+-CMs were also identified by Stereo-seq in adult mouse heart slices, with similar spatial expression profiles as a number of ventricular CCS markers such as *Hcn4*, *Cacna2d2*, *Cntn2*, and *Slit2* (Fig. S13b–d) supporting the persistence of *Dbh*+-CMs into the mature adult heart beyond the postnatal period.

## Genetic fate mapping and multiplexed nucleic acid in situ hybridization support that *Dbh*+-CMs, and *Dbh*+-derived CMs populate the developing CCS

To validate the transcriptomic identification of *Dbh*+-CMs, genetic fate mapping was conducted in developing and postnatal hearts. We developed three distinct genetic mouse models for such a purpose: *Dbh*Cre/Rosa26-tdTomato, *Dbh*-Knockin-CFP (*Dbh*CFP) and *Dbh*CreERT/Rosa26-tdTomato inducible lines. Whole embryos (E8.5, E9.5, E10.5,

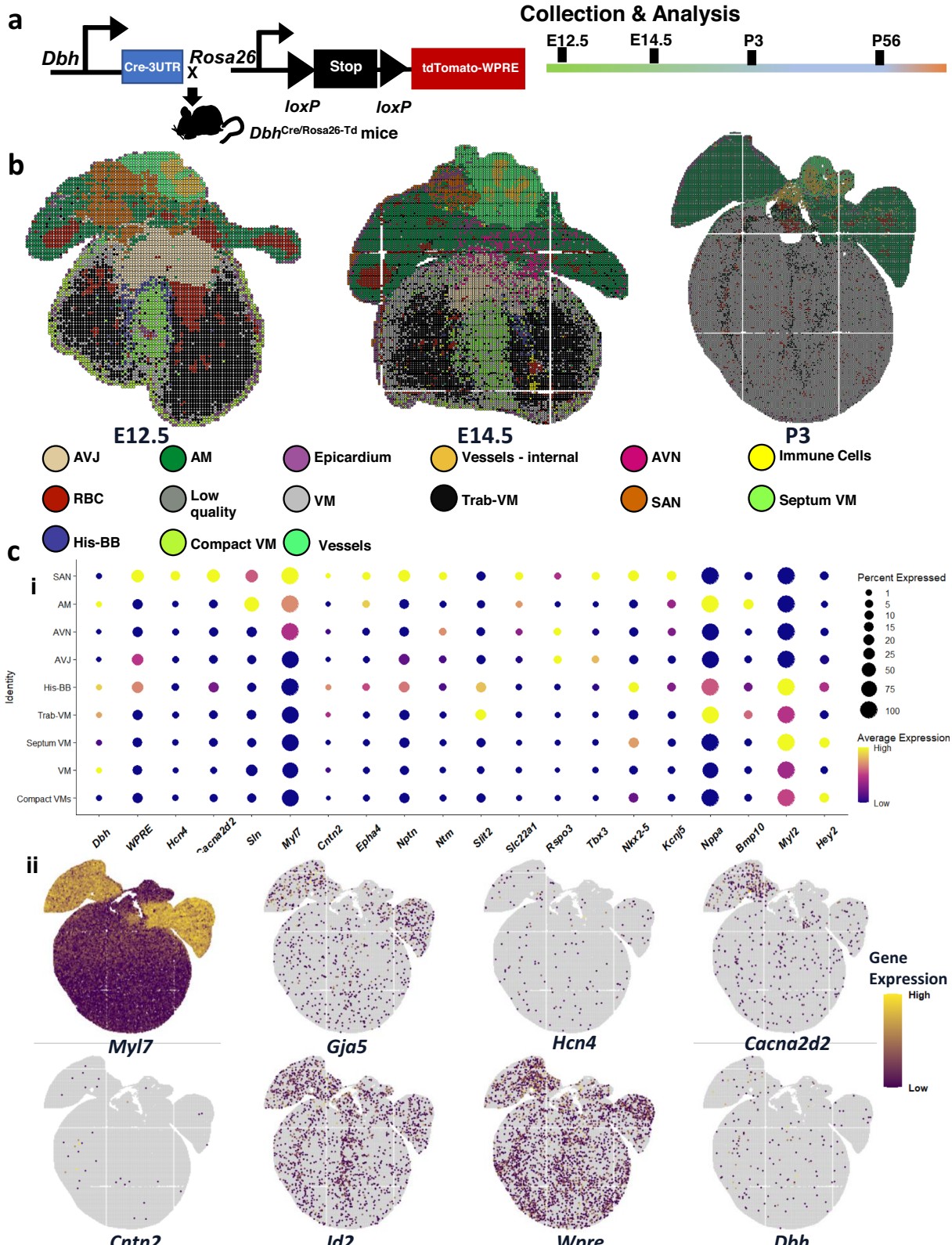

E12.5, E14.5, *n* = 5 embryos per stage) or isolated hearts (E12.5, E13.5, E14.5, E16.5, P3, P56, *n* = 5 hearts per stage) were analyzed by using multiplex nucleic acid in situ hybridization (RNAscope), immunohistological staining, and confocal microscopic imaging.

Genetic fate mapping was initially conducted on the *Dbh*^Cre/Rosa26-tdTomato mouse line to validate the findings from our Stereo-seq analysis on the same mouse line. The tdTomato fluorescence was

first observed in the dorsal neural tube at E10.5, likely reflecting neural expression in the developing CNS. From E10.5 to E12.5, tdTomato-expressing cells branched into groups migrating to the heart region via the pharyngeal arches (Fig. 3a). At E12.5, tdTomato fluorescence was observed in the sympathetic ganglion chain adjacent to the spinal cord, and, of particular interest, within the sinus venosus (SV, the primordial SAN) and right atrium (RA) (Fig. 3b). Thus, supporting the

**Fig. 2 | *Dbh* is expressed dynamically across the developing heart, as resolved with nanoscale-resolution Stereo-seq of a *Dbh*^Cre^/Rosa26-tdTomato reporter line. a** Schematic of *Dbh*^Cre^/Rosa26-tdTomato reporter line creation and analysis for spatially-resolved transcriptomics with Stereo-seq. **b** Plot of spatially-resolved transcriptomic pixels (each pixel consists of 20 × 20 from spots of individual diameters between 500 or 715 nm) identified across hearts from E12.5, E14.5, and P3 mice, coloured by cell type identified through unsupervised clustering with corresponding cell types coloured below. **c** (i) Dot plot describing *Dbh* and *Wpre* (a tdTomato construct-derived transcript) expression, with other key marker genes, across major cardiomyocyte populations identified through unsupervised clustering of E12.5, E14.5, and P3 Stereo-seq data. Dots are sized by percentage of cells

expressing each respective gene in each respective cell type. Dots are coloured by average expression of the respective gene in the respective cell type, with yellow indicating higher expression, and purple lower expression. (ii) Spatially-resolved plots describing *Dbh* and *Wpre* (a tdTomato-derived transcript) expression, with other key marker genes, across a slice from a P3 mouse heart sequenced with Stereo-seq. Yellow indicates higher expression and purple lower expression. VM ventricular myocardium, Trab-VM trabecular ventricular myocardium, SAN sinoatrial node, AVN atrioventricular node, AVJ atrioventricular junction, AM atrial myocardium, His-BB His-bundle branch, AM-CCS atrial cardiac conduction system, PKJ Purkinje fibres, VM-CCS ventricular cardiac conduction system.

population of the developing CCS with *Dbh*^+^-derived CMs, as denoted by tdTomato fluorescence at the protein level within the primordial CCS.

To determine the relationship between *Dbh*^+^ cells and the cardiac conduction system, RNAscope was applied on embryonic hearts at E12.5 (Fig. 3c, d). Bmp10 probe was applied to label the RA and trabecular regions (Fig. 3c). The expression and colocalization of *Dbh* and CCS markers *Cacna2d2*, *Hcn4*, *Id2*, *Shox2*, *Tbx18* were investigated by co-staining by multiplexed RNAscope in wild-type murine hearts from E12.5 to postnatal stages (Fig. 3 and Fig. S16,17). At E12.5, *Dbh* is abundant in the SV, HIS and PKJ and are well overlapped with *Cacna2d2* (Fig. 3d). At this stage, the AVN region marked by *Id*2 expression demonstrates a co-expression of *Id*2 and *Dbh* but *Dbh* has less abundant signal compared with Id2 expression. (Fig. 3d and Fig. S16). At E14.5, *Cacna2d2*, *Id2*, *Shox2*, *Tbx18* and *Hcn4* are overlapped with *Dbh* in various degrees in different parts of CCS, illustrating the existence of *Dbh*^+^-cells in CCS regions at this stage (Fig. S17a, b). Especially, abundant *Dbh* is observed in the SAN and RA regions where *Cacna2d2, Id2* and *Hcn4* signals marked, shown in Fig. S17b. At P3, notable *Dbh* signals were observed in SAN and both left and right PKJ, together with CCS marker *Cacna2d2* shown in Fig. S17c. Taken together, these findings strongly suggest that *Dbh*^+^ cells are potentially involved in CCS development and function.

The cell type identity and localization of *Dbh*^+^-derived cells and *Dbh*^+^-cells were further determined and delineated at E14.5, postnatal (P3) and adult (P56) stages. *Dbh*^Cre^/Rosa26-tdTomato mouse line labelled historically expressed *Dbh*^+^-cells as tdTomato fluorescence. We found that tdTomato-expressing cells were highly enriched in the cardiac conduction system and ventricular trabecular regions throughout these stages (*n* = 5 hearts examined per stage). More specifically, tdTomato-expressing cells were detected in the AVN and His bundle regions (Fig. 4a–c). Furthermore, some tdTomato expressing cells were also detected in the ventricular trabecular region (Fig. 4a) across the heart development, which is consistent with the developmental origin of PKJ[5,6]. By staining with the myocyte marker α-Actinin, we further confirm that the majority of tdTomato-expressing cells are cardiomyocytes, thus further confirming the identity of the cells as *Dbh*^+^-derived CMs (Fig. 4a–d).

To further elucidate spatially anatomical characteristics and expression pattern of *Dbh*^+^-derived CMs in postnatal and adult hearts, we interrogated the hearts with imaging and 3D computational image reconstruction. We characterized the spatial distribution of *Dbh*^+^-derived CMs across the whole heart and their structural relationship with specific components of the myocardium in neonatal (P3) and adult (P56) hearts (2-month-old) using *Dbh*^Cre^/Rosa26-tdTomato mouse line. Based on a series of microscopic histological images from P3 and adult hearts, we developed three-dimensional (3D) computational image reconstruction to determine the spatial distribution of *Dbh*^+^-derived CMs in P3 (Fig. 4e) and adult (Fig. 4f) hearts. The 3D models revealed spatial distribution characteristics of *Dbh*^+^-derived CMs in CCS (i.e., AVN, and His-Purkinje network) as shown in Fig. 4e, f, consistent with our earlier Stereo-seq analyses shown in Fig. 2d. Multi-angle view of the 3D reconstructions for such spatial distribution of

*Dbh*^+^-derived CMs in P3 and adult hearts are also illustrated in videos S1 and S2. From E14.5 to adulthood (Fig. S18a, b), *Dbh*^+^-derived CMs were abundant in the CCS regions where sympathetic innervation is enriched, as detected by immunostaining with anti-Th antibody, particularly in the adult heart (Fig. S18b).

## The spatial distribution of *Dbh*^+^-CMs is closely associated with the ventricular CCS

We then developed a *Dbh*^CFP^ mouse line to further reveal the expression and localisation of *Dbh* expressing cells (*Dbh*^+^-cells) in the developing and adult heart. Due to the weak genetic CFP signal, anti-Flag or anti-CFP antibodies were applied to enhance the CFP signal (defined by expressing CFP). As shown in Fig. 5a, distinct *Dbh*^+^-CMs, confirmed with cardiomyocyte marker α-Actinin staining, were detected in E14.5 and adult (P56), mostly abundant in AVN and ventricular CCS. We further established *Dbh*^CreERT^/Rosa26-tdTomato inducible reporter mouse line that allows us to map the *Dbh*^+^-CMs under a temporally controlled manner. Adult mice were treated at week 8 with 5-day tamoxifen administration and dissected after one-week post-induction. The tdTomato fluorescence was observed especially in AVN and HIS regions of CCS (Fig. 5b) as cardiomyocyte identity validated by cardiomyocyte marker α-Actinin staining, consistently with our results obtained in *Dbh*^Cre^/Rosa26-tdTomato and *Dbh*^CFP^ mice lines (Figs. 4d, 5a), where sympathetic innervation is enriched, as detected by Th positive cells exemplified in Fig. S19.

To further examine if *Dbh*^+^-derived CMs contribute to PKJ network, we then established mouse model *Dbh*^Cre^/Rosa26-tdTomato/ Cx40-eGFP by crossing a *Dbh*^Cre^/Rosa26-tdTomato reporter line with Cx40 transgenic eGFP (Cx40-eGFP) line that demarcates the PKJ network. A clear co-localization of tdTomato and eGFP signals was observed in both left and right ventricular Purkinje fibre networks (Fig. 5c) in *Dbh*^Cre^/Rosa26-tdTomato/Cx40-eGFP neonatal hearts (*n* = 5 hearts examined). Notably, the *Dbh*^+^-derived CMs (tdTomato-expressing cells) are more extensive than eGFP^+^-cells in the ventricular endomyocardium. The result suggests that *Dbh*^+^-derived CMs form cardiomyocyte populations other than Purkinje conductive cells, which is in agreement with our earlier transcriptomic analyses (Fig. 1c, Fig. 2c, d, Fig. 4c).

In summary, by using three mouse lineage tracing models and various techniques, despite some minor differences in signal distribution due to the different technologies used, we can consistently validate the identity of the majority *Dbh*^+^-CMs and *Dbh*^+^-derived CMs in the developing, postnatal and mature hearts. Their unique localization in CCS and co-localized with CCS markers are uncovered and verified, strongly suggesting their physiological role in the CCS, especially in HIS and PKJ where *Dbh*^+^-derived CMs are found to co-localised with Cx40^+^-cells.

## Electrophysiological characterization of *Dbh*^+^-CMs and *Dbh*^+^-derived CMs indicates they form a functional part of the CCS

To elucidate the physiological function of *Dbh*^+^-CMs and *Dbh*^+^-derived CMs in adult hearts, we have performed two series of experiments by interrogating the hearts with optogenetic electrophysiology or in vivo

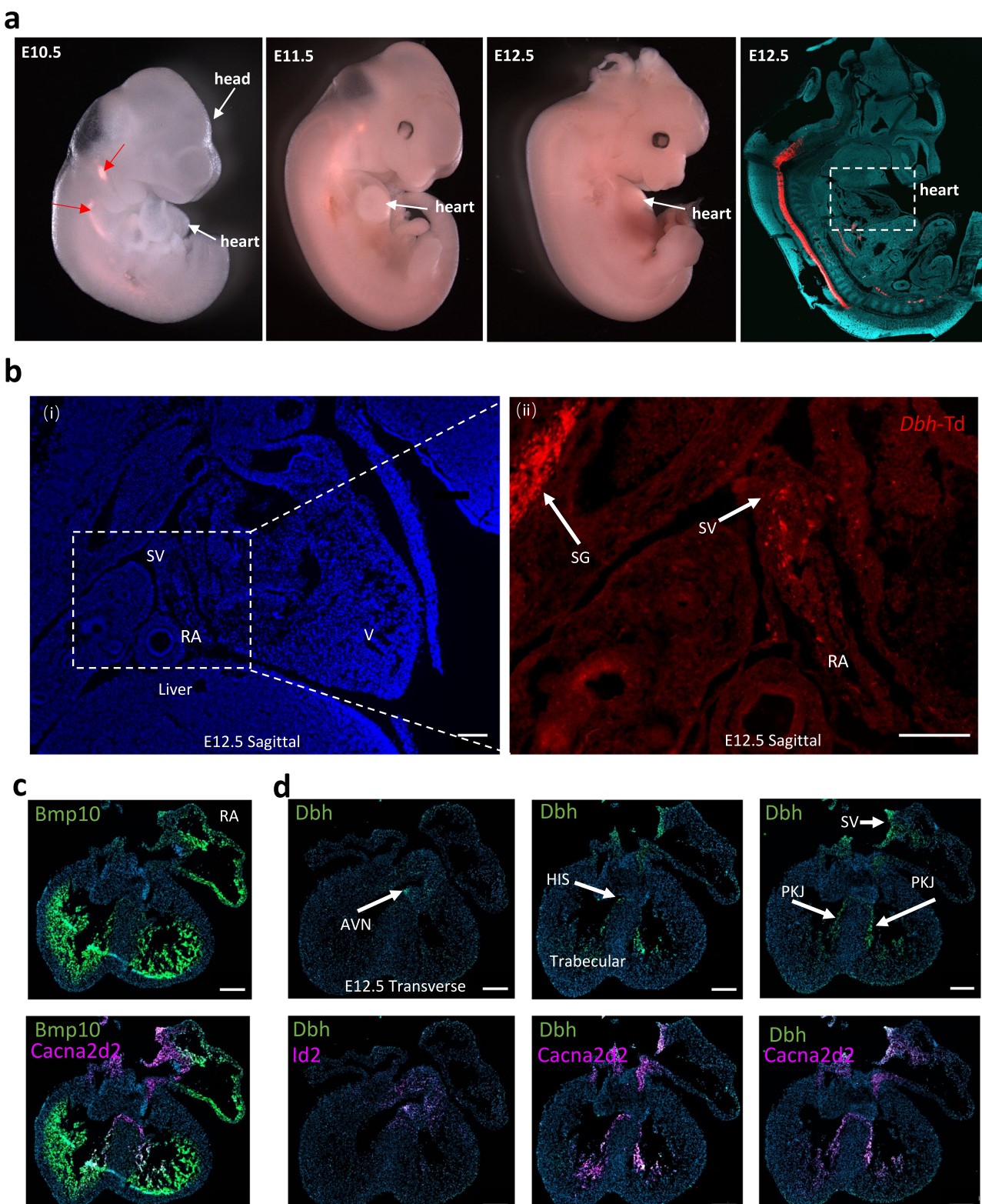

**Fig. 3 | Prospective fate mapping analysis of *Dbh^Cre*/Rosa26-tdTomato heart throughout the developmental stages. a** Representative gross images of tdTomato-expressing signals from E10.5 to E12.5 mouse embryos (left 3 panels) and an image of tdTomato-expressing signals of a sagittal section of a E12.5 embryo (far right panel). **b** Representative images showing tdTomato signals in RA and SV region of two sagittal section of E12.5 embryos (i) and magnified RA area (ii). **c** Representative RNAscope images showing the distribution of Bmp10 (green), CCS markers Cacna2d2 (magenta) in a whole field of E12.5 heart, indicating the anatomy of right atria, trabecular region and main CCS regions. **d** Representative RNAscope images showing the distribution of Dbh (green), CCS markers Id2(magenta) and Cacna2d2 (magenta) in whole field images of E12.5 heart respectively, unveiling the overlap expression in AVN, SV HIS and PKJ. Scale bar: 200 μm. Red Arrowed: tdTomato-expressing cells SG Sympathetic Ganglia, SV Sinus Venosus, RA Right Atrium, V Ventricle, AVN Atrioventricular nodal, HIS His-bundle branch, PKJ Purkinje fibre network, *n* > =3 hearts/slices for each stage.

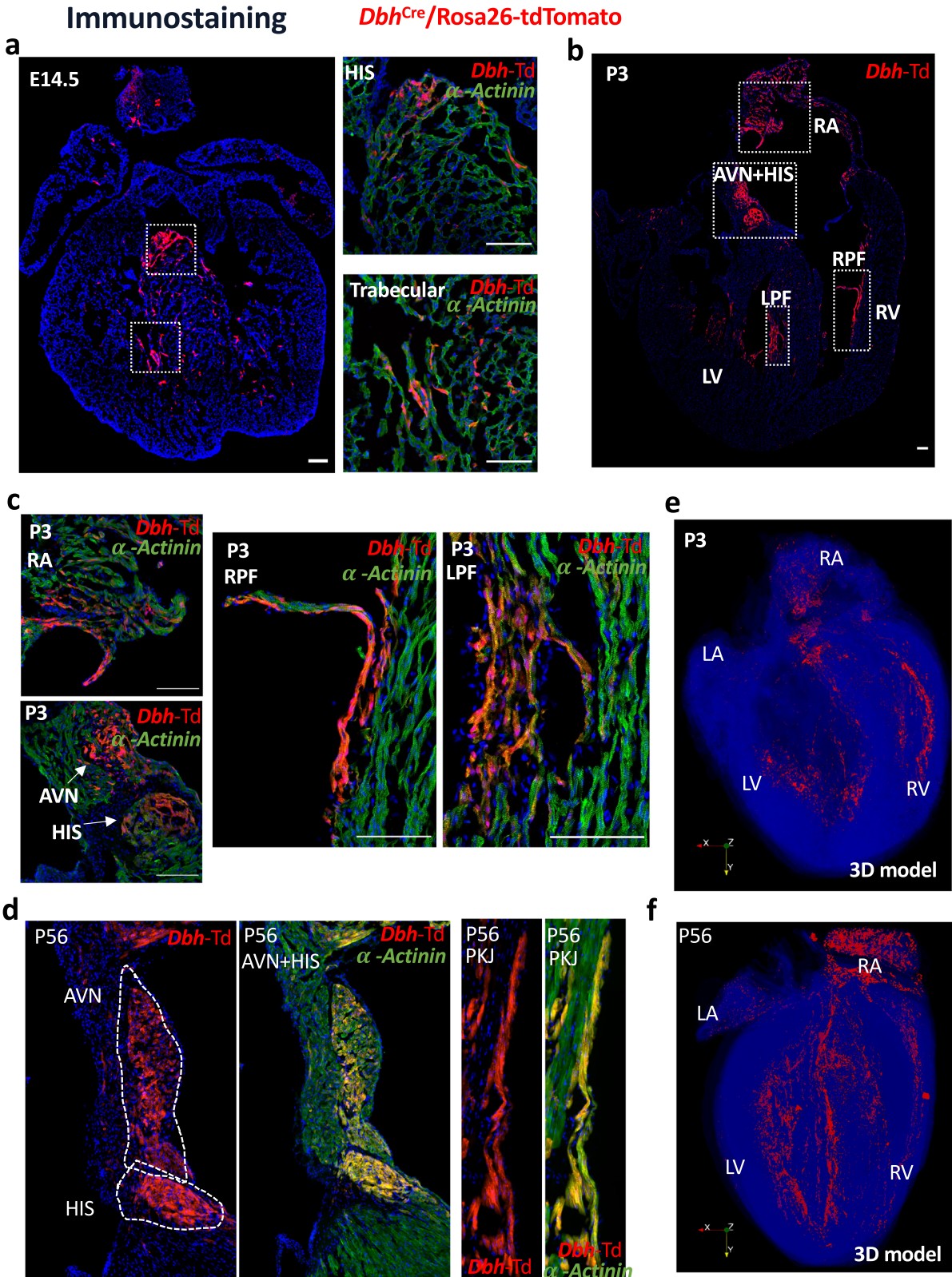

and ex vivo cardiac electrophysiological investigations of the mice with cardiomyocyte specific deletion of Dbh (*Dbh*cko) in comparison with their control littermates (*Dbh*f/f).

Firstly, we determined if *Dbh*+-derived CMs are associated with ventricular CCS, we compared photostimulation-induced electrophysiological characteristics of *Dbh*Cre/ChR2-tdTomato hearts with *Cx40*CreERT/ChR2-tdTomato and *MHC*Cre/ChR2-tdTomato hearts.

*Cx40*CreERT/ChR2-tdTomato mice have been used as an optogenetic tool for studying ventricular CCS by expressing ChR2 in Purkinje fibres[28], while *MHC*Cre/ChR2-tdTomato line has been used for studying cardiomyocytes in general as a non-selective cardiomyocyte ChR2 expressing mouse model. Despite such technique provide specific focal epicardial photostimulation to determine cell type dependent responses, due to tight electronic connection between ChR2

**Fig. 4 | Fate mapping of *Dbh*⁺-derived cardiomyocytes in the developing and postnatal hearts by using *Dbh*^Cre^/Rosa26-tdTomato reporter. a** Representative images of immunostaining showing the co-expression of tdtomato (red) and α-actinin (green) illustrating the distribution of *Dbh*⁺-derived CMs in HIS and trabecular regions at E14.5. Scale bar: 100 μm. **b** Representative image of tdTomato expressing cells in a four-chamber view section of P3 heart. Scale bar: 100 μm. **c** Representative images of immunostaining showing the co-expression of tdTomato (red) and α-actinin (green), illustrating the distribution of *Dbh*⁺-derived cardiomyocytes in RA, HIS, LPF and RPF regions in a heart at P3. Scale bar: 50 μm. **d** Representative images of immunostaining showing the co-expression of tdTomato (red) and α-actinin (green) illustrating the distribution of *Dbh*⁺-derived cardiomyocytes in AVN, HIS, PKJ regions in an adult heart. Scale bar: 100 μm, 100 μm, 10 μm, 10 μm (From left to right). **e** The distribution profile of *Dbh*⁺-derived CMs and multi-angle hyperspectral three-dimensional reconstruction results for *Dbh*⁺-derived CMs in a P3 heart sections. All results were written into VTK file format. **f** The distribution profile of *Dbh*⁺-derived CMs and multi-angle hyperspectral three-dimensional reconstruction results for *Dbh*⁺-derived CMs in adult (P56) heart sections. All results were written into VTK file format. AVN atrioventricular node, PKJ Purkinje fibre network, HIS His-bundle branch, LV Left ventricular, RV Right ventricular, RPF Right Purkinje fibre network, LPF Left Purkinje fibre network, RA Right atria *n* = 3 hearts/slices for all stages.

expressed cells and non-ChR2 expressing cells, such cell-type dependent responses may not be accurate. To enable timely and spatially controlled photostimulation of ChR2 hearts, we used a fibre optic delivering 470 nm light pulses (5 ms), generated by a time-controlled light emitting diode (LED) directed towards the epicardium of Langendorff-perfused hearts in LA, RA, LV and RV regions (Fig. 6a). ECGs were then recorded for electrophysiological analysis. We compared the regional responsiveness to photostimulation and ECG waveform morphologies of Langendorff-perfused hearts from three lines measured at sinus rhythm or light pacing at cycle length 100–120 ms. MHC^Cre^/ChR2-tdTomato (MHC-ChR2) hearts show uniform responsiveness to photostimulation with blue light in all four chambers (LA, RA, LV, RV) of the heart, and all 6 subregions of the heart, with each ventricle separated into its base and apex (Fig. 6a). However, Dbh^Cre^/ChR2-tdTomato (Dbh-ChR2) and ^Cre^/ChR2-tdTomato (Cx40-ChR2) hearts predominantly respond to photostimulation of the atria and the basal RV (Fig. 6a). However, (Dbh-ChR2)and Cx40-ChR2 hearts often show wider and variable QRS waveforms than that of MHC-ChR2 hearts. This demonstrates that Dbh-ChR2 hearts have a similar photostimulation response to Cx40-ChR2 hearts, in contrast to MHC-ChR2 hearts, in terms of chamber-specificity and RV light pacing (LP)-induced QRS waveform characteristics. In contrast to the uniform response to photoactivation of the α-MyHC-ChR2 ventricles, Purkinje fibre stimulation using the *Cx40*^Cre^/ChR2 line yielded variable electrical response depending on the illumination site, particularly ectopies triggered by photoactivation of the atrio-ventricular bundle, had QRS duration identical to the spontaneous complex, which is consistent with the physiological characteristics of the different conduction system regions.

We then characterised and compared the RV effective refractory periods (ERPs) determined by RV epicardial optical programmed pacing S1S2 protocol selectively photostimulation of *Dbh*-ChR2, Cx40-ChR2 and MHC-ChR2 expressing cells in these models. As summarised in Fig. 6b, RV ERPs are significantly longer in both *Dbh*-ChR2 and Cx40-ChR2 hearts than those of MHC-ChR2 hearts (Fig. 6c), the specific longer RV ERPs observed in Dbh-ChR2 and Cx40-ChR2 hearts indicate the association of *Dbh*⁺-derived CMs and Cx40⁺-derived cardiomyocytes with Purkinje fibres in RV as that reported previously in Cx40^CreERT^/ChR2-tdTomato mice[28].

Finally, to further interrogate the specific association of *Dbh*⁺-derived CMs with RV Purkinje fibre network, we applied iodine/potassium iodide solution (i.e. Lugol's solution), a well-recognised approach used for chemical ablation of ventricular Purkinje fibres[28,29]. Regardless of the photoactivation site or intensity, the response to RV photoactivation of Dbh-ChR2 and Cx40-ChR2 hearts, but not MHC-ChR2 hearts, was abolished by intracavital injection of Lugol's solution for 5 mins as exampled in Fig. 6d. Lugol's solution treatment caused prolongation of the QRS complex and abolished light-induced ectopies in Dbh-ChR2 and Cx40-ChR2, but not MHC-ChR2 hearts.

Taken together, the photostimulation-induced electrophysiological characteristics of *Dbh*-ChR2 and Cx40-ChR2 hearts are fundamentally similar, chemical ablation of ventricular Purkinje fibres by Lugol's solution treatment further proves the association of the *Dbh*⁺-derived CM with Purkinje fibres, similar to the association of Cx40 derived myocytes with Purkinje fibres.

To further gain the mechanistic insights of *Dbh*⁺ cardiomyocytes in cardiac physiology, we developed *Dbh*^f/f^ and *Dbh*^cko^ mouse models. *Dbh*^cko^ mice with selective cardiomyocyte *Dbh* deletion provide a valuable model to determine the unique function of *Dbh*⁺ cardiomyocytes. We conducted both in vivo and ex vivo electrophysiological and functional characterizations. As shown in Fig. 7 and Fig. S20. At baseline, *Dbh*^cko^ mice (*n* = 6) show comparable ECG and echocardiographic characteristics to their control littermates *Dbh*^f/f^ mice (*n* = 6), indicating that cardiomyocyte specific deletion of *Dbh* gene does not affect cardiac electrical and mechanical function at baseline condition in vivo. However, isolated Langendorff-perfused *Dbh*^cko^ hearts (*n* = 6) have significantly longer P-R intervals, AVN effective refractive periods (AVNERP), and atrial-ventricular (A-V) conduction Wenckebach block periods than that of *Dbh*^f/f^ hearts (*n* = 6). The results provide the first evidence that cardiomyocyte specific deletion of *Dbh* gene affects CCS electrophysiological properties.

## Characterization of catecholaminergic properties of *Dbh*⁺-CMs

To determine whether *Dbh* expression in *Dbh*⁺-CMs is associated with any function ex vivo, we conducted transmission electron microscopy (TEM) imaging and energy dispersive X-ray spectroscopy (EDS) analysis of chromium signal in comparison with adrenal medullary chromaffin cells as a positive control. Dissected tdTomato-expressing heart tissue and adrenal medullary tissue from *Dbh*^Cre^/R26-tdTomato mice were fixed and stained with chromium staining agents and imaged. Chromium staining is specific for staining catecholamine granules (vesicles), as when fixed with chrome salts, catecholamine granules are oxidized to a specific chromium signal that can be detected and quantified by EDS. As illustrated in Fig. 8a, b, cardiomyocytes from the *Dbh*⁺-CMs-rich atrioventricular junction, in particular, the pacemaker type cell as shown in Fig. 8a, contain high electron density vesicles with chromium signal as seen in chromaffin cells of adrenal medullary tissue (Fig. 8c), albeit these are fewer in number and a weaker chromium signal in cardiomyocytes compared with chromaffin cells as exemplified in Fig. 8d.

## Discussion

By utilising advanced and complementary transcriptomic analyses, genetic fate mapping, optogenetics, imaging and 3D computational image reconstruction, our study has interrogated developing and adult murine hearts, providing new insights into the development and physiology of the mammalian CCS through discovery and characterisation of novel *Dbh*⁺-CMs and *Dbh*⁺-derived CMs populations. There are several notable findings from our study:

Firstly, through scRNA-Seq with meaningful cardiomyocyte-focused quality control and utilizing a novel local and global structure-preserving dimensionality reduction technique (PHATE); we were able to discriminate rarer cardiac cell types, such as the CCS. We were left with some particular populations that were hard to classify, such as the AM-CCS. For example, AM-CCS was close in 3D PHATE space to other CCS populations (Data S2) and had expression of both atrial-like

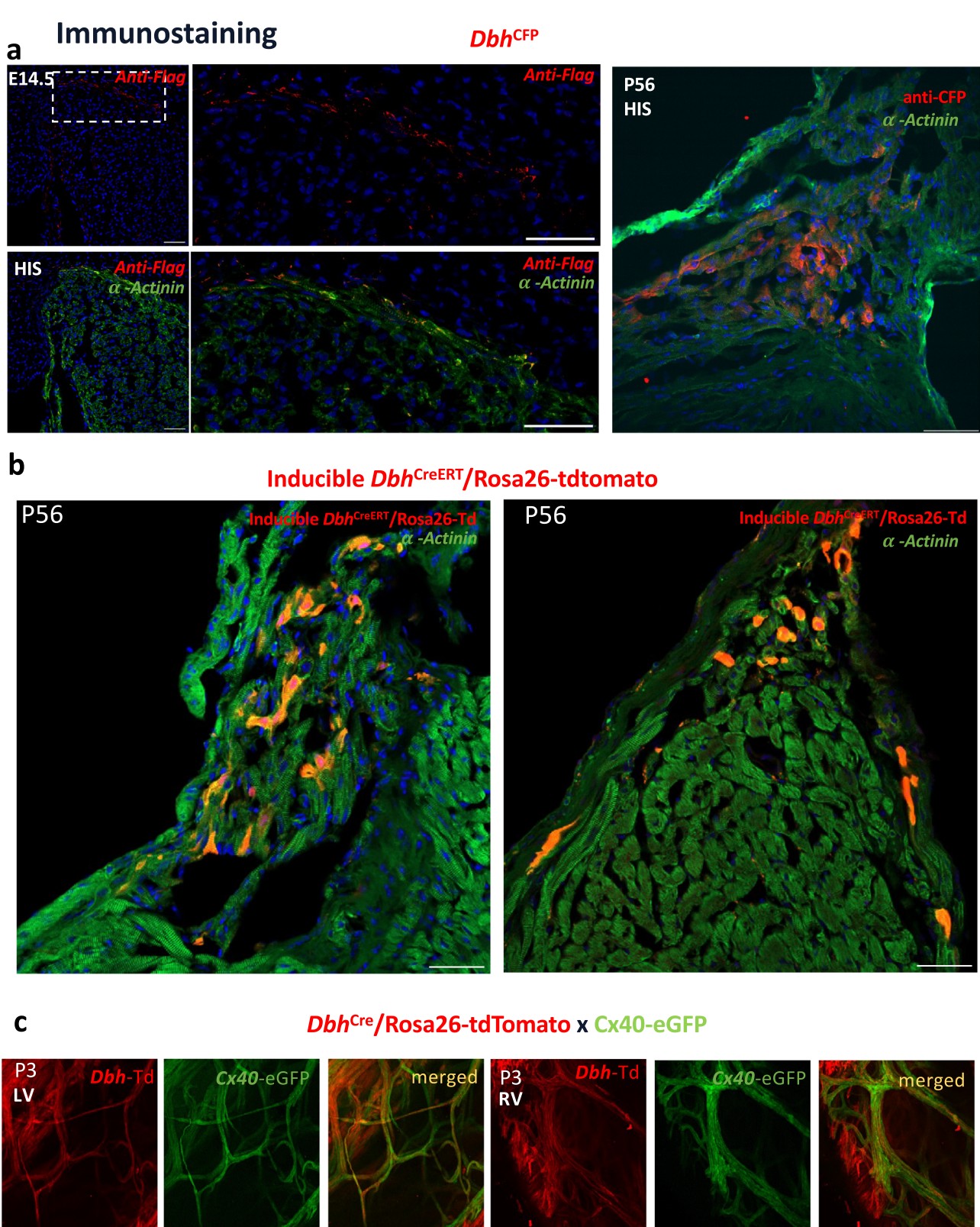

**Fig. 5 | Validation of spatial distribution of *Dbh*⁺-CMs and their close associa-
tion with CCS. a** Representative images of immunostaining with anti-Flag (red) and
anti-CFP (red), plus α-actinin (green) antibodies showing the expression of *Dbh*⁺-
CMs in HIS region at E14.5 and adult (P56) by using *Dbh*^CFP reporter.
**b** Representative images in AVN and HIS with α-actinin (green) immunostaining
showing the expression pattern of *Dbh*⁺-CMs by using *Dbh*^CreERT/Rosa26-tdTomato
inducible reporter at P56. **c** Representative images showing co-localization of *Dbh*⁺-
derived CMs and Cx40⁺ cells in PKJ network in both left and right ventricle in *Dbh*^Cre/
ChR2-tdTomato/Cx40-eGFP neonatal mouse hearts. Scale bar: 50 μm *n* = 3 hearts/
slices for each stage.

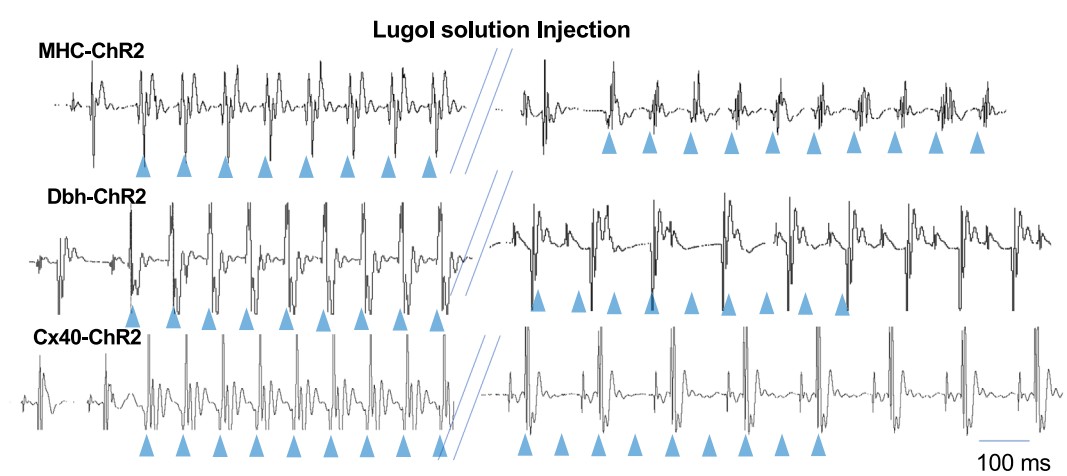

markers (*Myl7hi*, *Myl4hi*) and CCS-associated markers (Fig. S9). We believe the AM-CCS group to likely represent either some transition zone element of the SAN, or perhaps interatrial conduction pathways. AM-CCS had low-level expression of both canonical and novel SAN markers, such as *Shox2*, *Smoc2*, *Hcn4*, and *Pcdh17*. However, AM-CCS had high expression of genes associated with the transition zone of the SAN such as *Dkk3* and *Scn5a*[24] and with other CCS components such as

*Epha4*, *Nptn*, and *Gja1*[24,30]. However, it remains for future work to further characterize and validate the transcriptomic, proteomic, and physiomic signature of this population and others identified in this work.

Previous work has produced transcriptomic characterisations of the developing heart both across the whole organ and with specific focuses, such as on the cardiac conduction system. These works

**Fig. 6 | Direct optogenetic assessment of Purkinje fibre function implicates Dbh⁺-derived CMs in the ventricular CCS. a** Representative ECG traces from Langendorff-perfused hearts subjected to localized light pacing in four different chambers and 6 different subregions of hearts (SR: sinus rhythm, LA: left atrium, RA: right atrium, RVbase: right ventricle base, LVbase: left ventricle base, RVapex: right ventricle apex, LVapex: left ventricle apex) with blue light isolated from optogenetic transgenic mouse lines expressing the fused protein tdTomato-ChR2 under the control of the promoters for MHC (MHC-ChR2), Dbh (Dbh-ChR2) and Cx40 (Cx40-ChR2). Blue triangles indicate when blue light pacing was applied. **b** A diagram displaying the electrical responsiveness of the different heart chambers to similar regional optical pacing across the different mouse genetic lines. **c** Comparison of effective refractory periods (ERP) determined by RV epicardial

optical programmed stimulation in hearts from three optogenetic transgenic mouse lines. ($n = 5$ for MHC-ChR2 model; $n = 7$ for Dbh-ChR2 model; $n = 5$ for Cx40-ChR2 model). **d** Representative ECG traces of ectopic beats originated by epicardial light stimulation of the RV from three optogenetic transgenic mouse lines before and after (5 min) RV intracavital Lugol's solution injection ($n = 5$ mice per mouse line). Blue arrows indicate the light pulses. Lugol's solution treatment caused enlargement of the QRS complex and abolished light-induced ectopies in Dbh-ChR2 and Cx40-ChR2, but not MHC-ChR2 hearts. Both genders of mice aged 8–12 weeks are utilized in ex vivo ECG studies. Statistical results analysis was performed using GraphPad Prism 8, and statistical significance was calculated by a one-way ANOVA test.

provide a basis for understanding a range of biological processes in cardiac development, such as: early cardiac mesoderm commitment[20,31], cross-species differences[32,33], and molecular markers of different anatomical regions of the CCS[34].

We sought to identify pertinent catecholamine-associated populations of heart cells, based on prior work by ourselves and others suggesting a connection[34–37]. In contrast to the pioneering work of others producing extensive mapping of known cell types in transcriptomic detail;[34] we sought to use whole-organ and multi-stage scRNAseq and spatial transcriptomics for unbiased hypothesis generation and cross-validation in identifying previously unknown cell types. We were not able to pre-suppose at what single stage of development a snapshot would be appropriate to identify these cells. Thus, we performed multi-stage sequencing from E8.5 to P56, across droplet-based scRNAseq and large-scale spatial transcriptomics. The use of droplet-based techniques enabled us to sequence many cells, but at the cost of lower sequencing depth than low-throughput well-based scRNAseq techniques. Whereas, the use of the novel large-field but nano-scale Stereo-seq, enabled us to embed our scRNAseq findings in anatomical ground truth and cross-validate our findings prior to embarking on resource-intensive structural and functional investigation of these catecholamine-associated cardiomyocytes. Our work brings novelty to provide a panoramic view from the early cardiac development through to adult hearts, across multiple sequencing technologies, with a focus in this paper on a specific question to find catecholamine-associated cardiomyocytes. However, we hope that our combined datasets will enable the wider scientific community to generate and investigate their own hypotheses, beyond our exciting presentation of this new putative cardiomyocyte cell type.

Secondly, by undertaking integrated scRNA-Seq and spatially resolved transcriptomic analyses of the developing heart, we established a comprehensive spatiotemporally-defined transcriptional landscape of cell types populating the developing murine heart. Our spatially resolved transcriptomic Stereo-seq investigation identified major cardiac cell populations and structures through both unsupervised and supervised machine learning techniques. However, we note that there were some imperfect, but logical, predictions of cell types through supervised techniques (Fig. S15). Integration of scRNA-Seq and spatially-resolved transcriptomics data remains a nascent field, and we expect future computational work to produce more accurate integration, perhaps through inclusion of prior knowledge and combinatorial scoring based on a cell's given cassette of initially predicted labels. The utilization of the nanoscale Stereo-seq technology with high resolution (centre-to-centre spots distance of 500 or 715 nm) and larger capture size provides the leading resolution and tissue capture area compared to other reported spatial transcriptomic techniques, with sequencing depth and breadth comparable to many single-cell sequencing techniques, but with spatial information to further advance understanding[17]. The adoption of these novel technologies integrated into a "traditional" biological workflow at the hardware and software level helps advance the integration of bioinformatics-led discovery, reducing resource consumption in

hypothesis generation and hypothesis 'pruning' prior to biological validation.

Thirdly, the most important, we identified novel cardiomyocyte populations, Dbh⁺-CMs and Dbh⁺-derived CMs and defined their association with the CCS in developing and mature murine hearts. Our investigation demonstrated that tdTomato reporter expression could first be detected in the neural tube at E10.5 and then in the pharyngeal arches between E10.5 to E12.5, which most likely reflects neural crest cell migration and their respective formation of sympathetic ganglia[38,39]. Our transcriptomic analyses indicate that although Dbh⁺-CMs represented only small percentages of the total working cardiomyocyte populations (Fig. S11), the CCS appeared to be over-represented in Dbh⁺ cardiomyocytes as percentages of their respective total cardiomyocyte lineage numbers. We are reassured that others have the Dbh signal present in their data from the developing and adult CCS, despite the fact we are the first to investigate further and characterize this signal. Others have utilized micro-dissection of the developing and adult CCS and observed, although not discussed, Dbh enrichment across the CCS, and in transcriptional programs associated with cardiac pacemaking tissue[24,25,40]. Our experiments demonstrate that Dbh⁺-CMs and Dbh⁺-derived CMs mainly populated the cardiac conduction system and ventricular trabecular regions throughout E14.5, and postnatal stages (Figs. 4, 5). Collectively, these results indicate that Dbh⁺-derived trabecular ventricular cardiomyocytes and Dbh⁺-derived ventricular PKJ myocytes likely share some common progenitor or transcriptional program, although we are unable to specify at what stage of development this progenitor or program arises or when any progeny commit to differential cell fates. We currently lack any evidence to prescribe any origin of Dbh⁺-CMs, or Dbh⁺-derived CMs, different to that of other cardiomyocytes. Further research is needed to understand the population of progenitors from which these cells arise.

Fourthly, we assayed the functional role of Dbh⁺-derived CMs by selectively stimulating these cells by expressing light-sensitive Channelrhodopsin-2 (ChR2) channels in these cells by crossing the Dbh^Cre line with the ChR2-tdTomato line or selectively inactivation of Dbh in cardiomyocytes by cardiomyocyte-specific (MHC^Cre/Dbh-flox) Dbh deletion (Dbh^cko). The genetic restriction of ChR2 expression allowed us to selectively interrogate the Dbh-derived cells as a functional part of the ventricular CCS by comparing them against a Cx40-ChR2 line, an established model for studying Purkinje fibre function[28]. In the alternative model used, the MHC promoter drives ChR2 expression to all cardiomyocyte types. Our results demonstrate that Dbh-ChR2 and Cx40-ChR2 hearts share remarkably similar photostimulation-induced electrophysiological characteristics and similar responses to intracavital Lugol's solution injection. Due to the higher thickness of the LV wall, Purkinje fibres failed to be photoactivated with epicardial photostimulation, as observed in previous work[28]. Our electrophysiological functional studies on Langendorff-perfused Dbh^cko hearts provide the first evidence that cardiomyocyte-specific deletion of Dbh affects CCS electrical function. Our in vivo study of intact Dbh^cko mice failed to reveal any difference from Dbh^f/f mice. The contrast

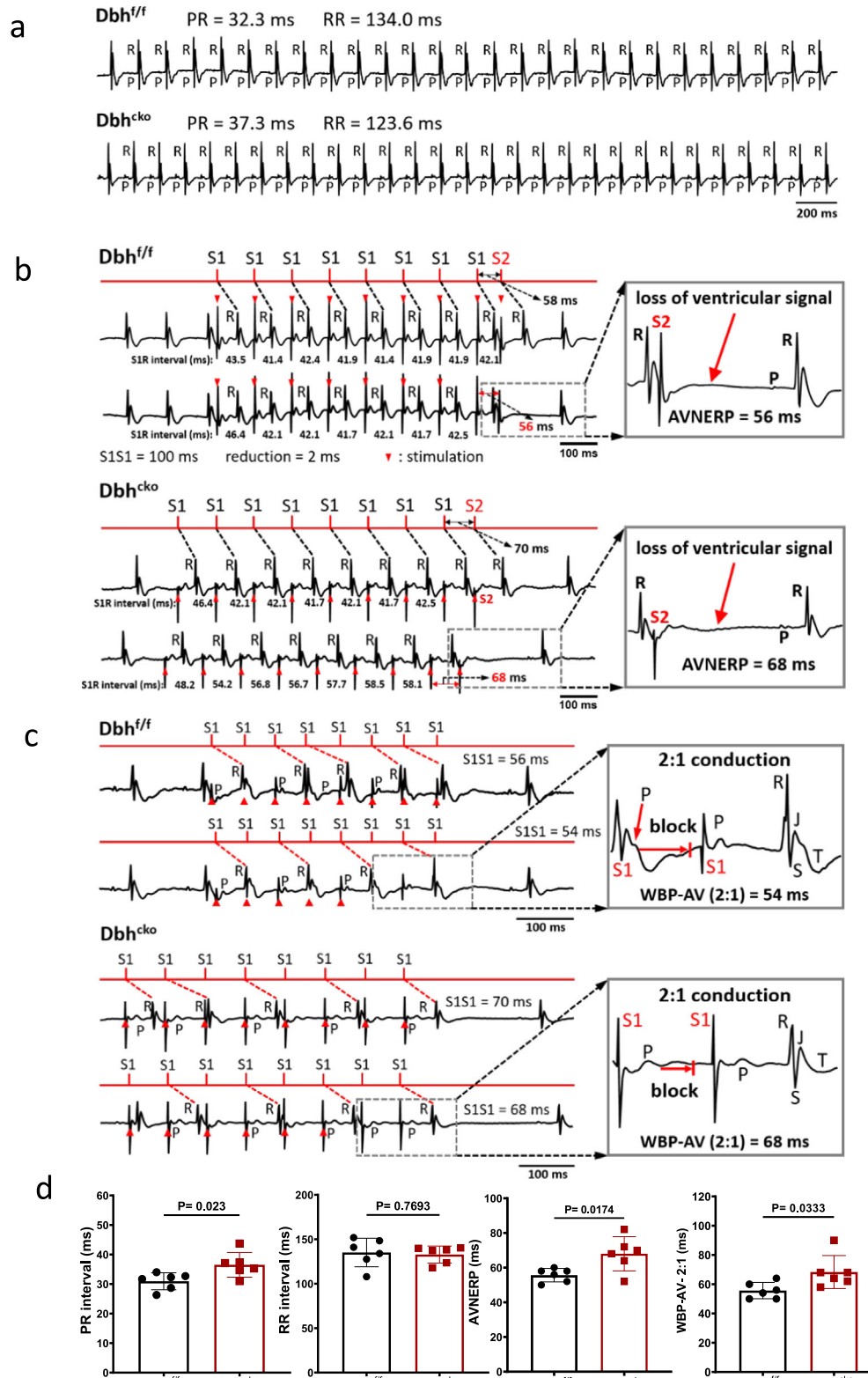

**Fig. 7 | Electrophysiological characterization of *Dbh*^cko and *Dbh*^f/f hearts reveals the role of cardiomyocyte *Dbh* expression in intracardiac conduction.** **a** Typical ECG traces of isolated hearts at baseline from *Dbh*^cko and *Dbh*^f/f mice. **b** Representative ECG traces for evaluation of AVNERP by programmed electrical stimulus (PES), the drop of S2 induced ventricular QRS complex is highlighted in the grey box indicating the AVNERP. **c** Consecutive S1S1 PES pacing protocol was conducted to measure the Wenckebach point at 2:1 conduction, the highlight part indicates the Wenckebach point—that only one QRS complex was generated by two S1 stimulations. **d** The statistical comparison of P-R, RR intervals, AVNERP and Wenckebach point between Dbh^cko and Dbh^f/f hearts. *n* = 6 per group. Both genders of mice aged 8–12 weeks are utilized in ex vivo ECG studies. Statistical results analysis was performed using GraphPad Prism 8, and statistical significance was calculated by a Student *t* test.

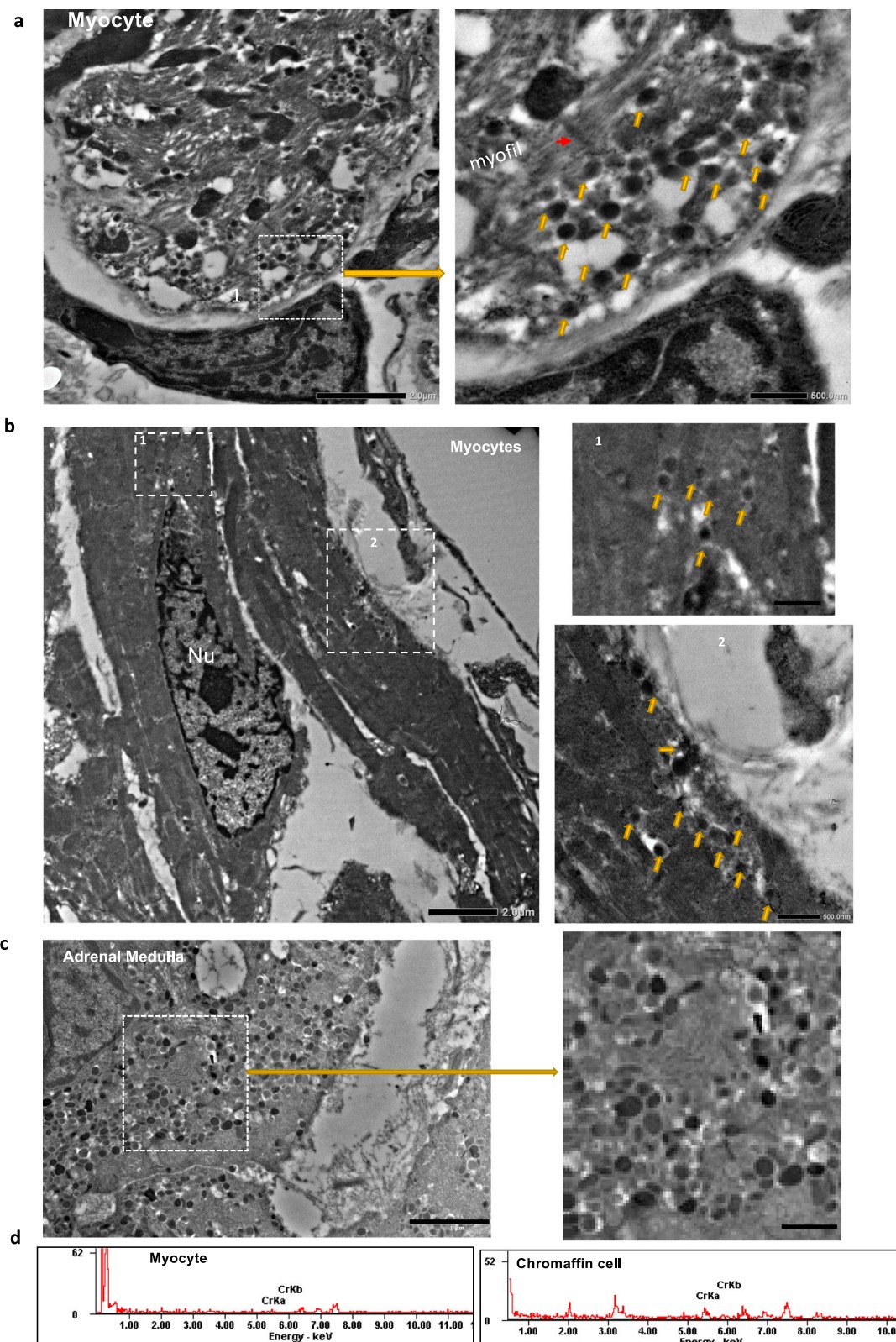

**Fig. 8 | Transmission electron microscopy identifies catecholaminergic vesicles within CMs. a**, **b** Transmission electron microscopy (TEM) imaging: high electron density vesicles (arrowed) detected in cardiac myocytes. **c** Transmission electron microscopy (TEM) imaging: high electron density vesicles (arrowed) detected in adrenal medullary chromaffin cells. **d** The selected region for EDS analysis of chromium. Weak chromium signals measured as CrKa and CrKb peaks in EDS spectrum were detected in cardiomyocyte in contrast to the strong chromium signals in adrenal medullary chromaffin cells. Nu, cell nuclei. Scales bar: 8b: left: 2 μm, right: 500 nm; 8c: left: 2 μm, right: 500 nm; 8d: left: 1 μm, right: 500 nm; $n = 3$ hearts.

between our isolated ex vivo heart and intact mice results suggests that the autonomic nervous system is likely able to account for the physiological deficit caused by *Dbh* deletion. The results suggest an important functional role of the cardiac Dbh system in modulating CCS function, but that it is not essential for normal rhythmogenesis. Our data thus strongly support that *Dbh*+-derived CMs form a functional part of the ventricular CCS and that *Dbh*+-CMs contribute most importantly to the atrio-ventricular junction.

Our study also provides the evidence of catecholaminergic-type vesicles containing adult cardiomyocytes from the *Dbh*+-CMs-rich atrioventricular junction. However, further investigations are required to ascertain how extensive these cells and their catecholaminergic functions are, and any influence they have on normal cardiac function or dysfunction. Others have previously reported cardiomyocytes may be involved in cardiac para-sympathetic nervous system signalling, through the canonical parasympathetic neurotransmitter acetylcholine[41]. The para-sympathetic and sympathetic nervous system are, canonically, antagonistic for regulating cardiac function. That the par-enchymal cells of the heart themselves may play a role in reg-ulating the balance between stimulatory and suppressive signalling independent of higher neural control, seems an attractive suggestion from an evolutionary perspective. Recently, evidence suggests that intercellular signalling between cardiac sympathetic neurons and cardiomyocytes may be channelled through elementary units, represented by the neuro-cardiac junctions "NCJs", where a 'quasi-synaptic' communication takes place, and whose properties fit well with those of neurogenic regulation of cardiac function[42,43]. Our data also show that *Dbh*+-CMs are abundant in areas enriched in sympathetic innervation. That *Dbh*+-CMs are enriched in the same anatomical regions (endomyocardially) in the ventricles supports that *Dbh*+-CMs are associated with the ventricular CCS, as sympathetic innervation of the ventricles is known to be restricted to the ventricular CCS. Furthermore, the close proximity of *Dbh*+-CMs and sympathetic varicosities opens a huge interest for future studies to elucidate the communication mechanism of the sympathetic nervous sys-tem and *Dbh*+-CMs and any physiological and pathophysiological implications.

*Dbh*+-cardiomyocytes have potential impact in physiology and pathophysiology beyond development. The sympathetic release of catecholamines, along with endocrine release from the adrenal glands, regulates the heart during changing demands in cardiac output, such as in exercise. However, cardiac catecholaminergic signalling is implicated in a range of chronic and acute disease states such as heart failure, myocardial infarctions, and sudden cardiac death[44–46]. Fur-thermore, there is specific evidence for the role of *Dbh* in both phy-siological regulation and disease. *Dbh*−/− mice display bradycardia, hypotension, and a failure to mount a cardiovascular response to exercise[47]. Humans with *DBH* deficiency also display orthostatic hypotension and in rare cases sudden cardiac death with CCS fibrosis[48]. Our findings from mouse hearts with cardiomyocyte-specific *Dbh* deletion provide the first experimental evidence that *Dbh*+-CMs may be important in physiological regulation of the CCS. There are a number of pathological conditions where the prospective pertinence of this new paradigm of catecholaminergic CMs warrants further investigation in new light, such as catecholaminergic polymorphic ventricular tachycardia, heart failure, atrial fibrillation, or post-orthotopic heart transplant. For example, it is interesting to spec-ulate whether these *Dbh*+-CMs might contribute to inter-individual variation in atrial fibrillation pathology through variable likelihood of passing atrial fibrillatory waves through to the ventricles. However, this remains for future work to investigate.

Given the gene-specific evidence, and the potential implications for wider catecholaminergic signalling; we sought to understand if our *Dbh*+-cardiomyocyte population persisted into adulthood, as this would be necessary for any implied involvement. We used Stereo-seq, RNAscope, immunostaining, multiple lineage tracing lines, and opto-genetics, to confirm that *Dbh*-derived cardiomyocytes were pre-dominantly in the CCS, and that active transcription of *Dbh* continued in the adult (P56) heart. Thus, we now have an additional piece of the puzzle that can be used to help understand not only these specific phenotypes in deficiency, but also the wider role of *Dbh*+-cardiomyo-cytes in physiology and pathophysiology in the adult.

Future work to further interrogate this novel *Dbh*+-cardiomyocyte population may support a growing change in mechanistic models of cardiac signalling. Adult cardiomyocytes already possess secretory potential, for natriuretic peptides as local and endocrine hormones[49]. It has already been identified that CMs express all the synthetic machinery for the production, storage, and release of Acetylcholine (ACh)[50,51], which is the main parasympathetic neurotransmitter. Addi-tionally, cardiomyocyte-specific ACh appears to be functionally rele-vant in maintenance of normal CM calcium handling[52,53]. These authors have suggested that these ACh-producing CMs might act to amplify the ACh signal from the parasympathetic neurons[50]. It is tempting to suggest that *Dbh*+-cardiomyocytes might occupy a similar role as ACh-producing CMs, especially given the overlap of *Dbh*+ cardiomyocytes rich areas of the heart with densely sympathetically innervated areas of the heart as we found. However, ACh is a simple molecular neuro-transmitter to synthesize, using readily available biosynthetic pre-cursors that are present irrespective of a specific cellular transcriptional programme. Thus, perhaps this increased complexity in the production of catecholamines mandates a different relationship than for ACh-producing CMs; where *Dbh*+-cardiomyocytes are much more reliant on the catecholamine precursor supply from other non-*Dbh*+ cardiomyocyte cellular sources, which needs to be explored in the future. If *Dbh*+-cardiomyocytes possess catecholamine uptake mechanisms, this could contribute to the local fine-tuning of the adrenergic signalling, and could be either through uptake, conversion to adrenaline, and subsequent release, perhaps emphasizing the receptor affinity differences between noradrenaline and adrenaline. Otherwise, the uptake and degradation of catecholamines could con-tribute to the regulation and termination of catecholaminergic sig-nalling locally as reported in adipose tissue[54]. The levels of complexity are such that it will take much effort, likely beyond any single colla-boration, to delineate the contribution of these exciting new cells.

Finally, our findings suggest a potential relationship between sympathetic innervation and *Dbh*+-derived CMs, during CCS forma-tion, or vice versa (Fig. S18). Figure 9 summarises our hypothetical relationships of sympathetic innervation and *Dbh*+-derived CMs and *Dbh*+ CMs contributed CCS formation in developing and mature hearts. It is known that sympathetic innervation occurs concurrently with the development of the cardiovascular system[55]. The importance of the sympathetic nervous system, or some other catecholaminergic source, is supported by the evidence of embryonic death with cardiac developmental defects due to two most important catecholamine synthetic enzymes, *Th*-deficient[10] and *Dbh*-deficient mice[11,12]. The rela-tionship between sympathetic innervation and cardiac pathology in the adult[56] and the perseverance of *Dbh*+-CMs into adult hearts, raises important pre-clinical questions about any putative interaction between the sympathetic innervation of the heart and *Dbh*+-CMs or *Dbh*+-derived CMs, which will be the future focus of our study.

In conclusion, we have studied the development and function of the murine cardiac conduction system by applying a sophisticated approach integrating single-cell RNA sequencing, high resolution spatially-resolved transcriptomics, unsupervised and supervised machine learning techniques, cell fate mapping, multi-modal imaging, and optogenetics. Our identification of *Dbh*+-CMs and *Dbh*+-derived CMs provide new insights into the mid-to-late development of the mammalian CCS through this previously unreported

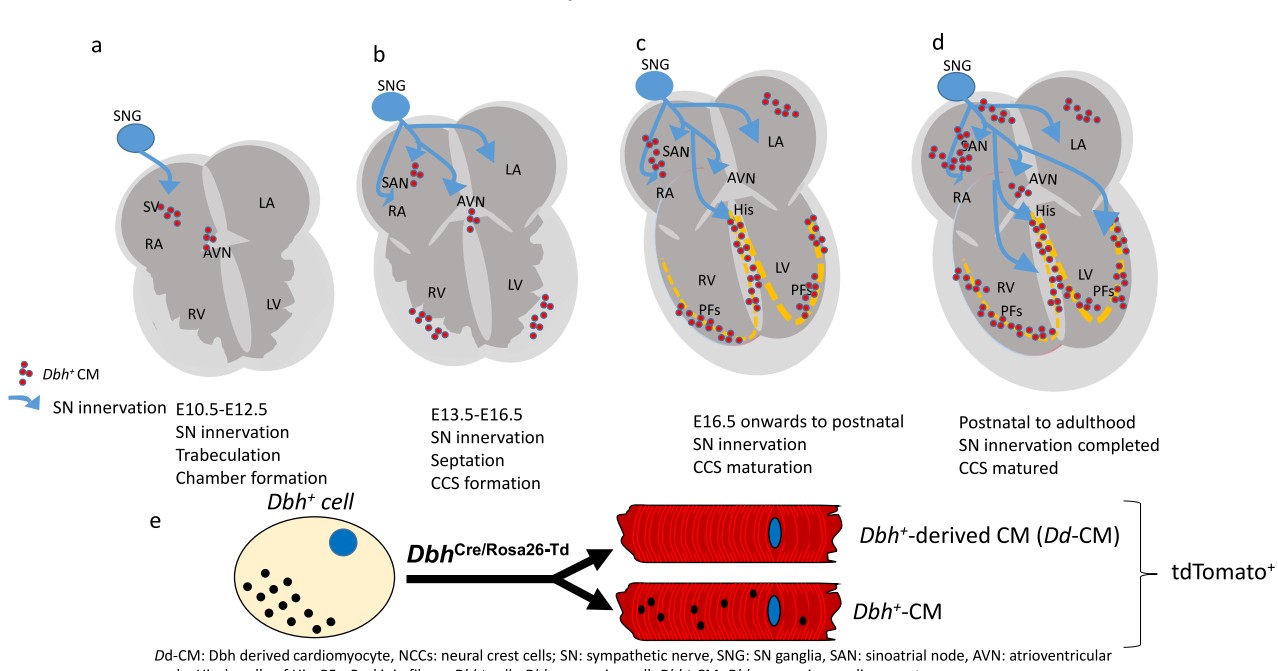

Contribution of *Dd*-CM to CCS formation

*Dd*-CM: Dbh derived cardiomyocyte, NCCs: neural crest cells; SN: sympathetic nerve, SNG: SN ganglia, SAN: sinoatrial node, AVN: atrioventricular node, His: bundle of His, PFs: Purkinje fibers, *Dbh*⁺ cell: *Dbh* expressing cell, *Dbh*⁺-CM: *Dbh* expressing cardiomyocyte

**Fig. 9 | Proposed relationships of SN innervation and *Dbh*⁺-CCS formation.**
**a** *Dbh*⁺-CM-associated CCS formation occurred concurrently with sympathetic innervation from E12.5 after neural crest cells migration occurred in SN ganglia and heart region from E10.5. Such process is in parallel to the processes of ventricular trabeculation and chamber formation. **b** From E14.5 onwards, the *Th* positive nerve fibres were clearly identified and they extended to the AV junction, His-Bundle and part of the ventricular myocardium. Such a process occurred concurrently with *Dbh*⁺-CM-associated CCS formation. **c** E16.5 onwards to postnatal, SN innervation continues and CCS is maturing. **d** From postnatal to adulthood, SN innervation and maturation of CCS are completed. **e** Illustrating the relationship between *Dbh*⁺ cells, *Dbh*-derived cardiomyocytes, and *Dbh*⁺-CMs: Dbh expressing cardiomyocytes.

catecholaminergic cardiomyocyte population. However, the function of these cells requires further studies in the future.

## Methods
No statistical methods were used to predetermine sample size. The experiments were not randomized and investigators were not blinded to allocation during experiments and outcome assessment.

### Animals and ethical approval
All animal experiments were performed on mice neonatal or adult mice (both genders) in accordance with the United Kingdom Animals (Scientific Procedures) Act 1986 and were approved by the University of Oxford Pharmacology ethical committee (approval ref. PPL: PP8557407) or Animal Care and Use Committee of the Southwest Medical University, Sichuan (China) (No: 20160930) in conformity with the national guidelines under which the institution operates. All mice including mutant mice and wild-type (WT) littermates used in this study were maintained in pathogen-free facilities at the University of Oxford or Southwest Medical University. Mice were given *ad libitum* access to food and water.

### Mouse models
The *Dbh*^Cre mouse model is generated by using CRISPR/Cas9 technique that the 2A-iCre-WPRE-polyA expression box is knocked in at the stop codon site of the *Dbh* gene through homologous recombination technology. The brief processes include: (1) Cas9 mRNA and gRNA are obtained by in vitro transcription; (2) homologous recombination vector is constructed by In-Fusion cloning method, which contains a 3.0 kb 5'homology arm, 2A-iCre-WPRE-polyA segment and 3.0 kb 3'homology arm; (3) Cas9 mRNA, gRNA and donor vectorwere microinjected into the fertilized eggs of C57BL/6 J mice to obtain F0 generation mice. The positive F0 generation mice identified by PCR

amplification and sequencing were mated with C57BL/6 J mice to obtain F1 generation mice.

*Dbh*^Cre/Rosa26-tdTomato mice were then obtained by crossing *Dbh*^Cre mice with B6.Cg Gt(ROSA)26Sortm9(CAG-tdTomato)Hze/J strain (Stock No. 007909, Jackson Labs) for lineage tracing study. *Dbh*^Cre/ChR2-tdTomato mice were obtained by crossing *Dbh*^Cre mice with B6.Cg-Gt (ROSA) 26Sortm27.1(CAG-COP4*H134R/tdTomato) Hze/J strain (Stock No. 012567, Jackson Labs). The resulting offspring have the STOP cassette deleted in cardiomyocytes, driving the expression of ChR2 [hChR2 (H134R)-tdTomato fusion protein]. *Dbh*^Cre mice were obtained from Shanghai Model Organisms Centre, Inc. (Shanghai, China).

*Dbh*^CreERT mouse model is generated by using CRISPR/Cas9 technique that the 2A-CreERT2-WPRE-polyA expression box is knocked in at the stop codon site of the *Dbh* gene through homologous recombination technology. The *Dbh*^CreERT/Rosa26-tdTomato mice was generated by crossing *Dbh*^CreERT mice with C57BL/6JSmoc-Gt(ROSA) 26Sorem(CAG-LSL-tdTomato)1Smoc mice from Shanghai Model Organisms Centre, Inc. (Shanghai, China).

*Dbh*^CFP mouse model is generated by using CRISPR/Cas9 technique that the IRES-CFP-5×Flag-WPRE-polyA expression box is knocked in at the stop codon site of the *Dbh* gene through homologous recombination technology by Shanghai Model Organisms Centre, Inc. (Shanghai, China).

*Dbh*^Cre/ChR2-tdTomato/Cx40-eGFP mice were obtained by crossed *Dbh*^Cre/ChR2-tdTomato heterozygous positive mice with a Cx40-eGFP mouse line[57].

The *Dbh*^f/f mice were homozygous *Dbh* flox mice with two *LoxP* sites flanking exon 3-5 of *Dbh* with CRISPR/Cas9 technique shown on Fig. S21a by Cyagen Biosciences (Suzhou) Inc. (Suzhou, China). Cardiomyocyte-specific *Dbh* knockout mice (*Dbh*^cko) were generated from the *Dbh*^f/f mice crossed with mice expressing cyclization

recombination enzyme (Cre) under α-myosin heavy chain (α-MHC) promoter. The genotypes for *Dbh*[f/f] and *Dbh*[cko] were verified with PCR. The efficiency of *Dbh* knockout were confirmed with Real-time PCR with the primer: Sense: 5′-GCTGGGGTCCTGTTTGGAAT-3′; Anti-sense: 5′ CTCCAGGCATCCGCAAAGT-3′(Fig. S21b, c).

Mice used for optogenetic electrophysiological study, transgenic adult male mice (8 weeks), with a genetic background C57B6J, expressing cre-recombinase under the control of either the MHC (B6.FVB-Tg(Myh6-cre)2182Mds/J, Stock No. 011038, Jackson Labs) or Cx40 promoter were bred with B6.Cg-Gt (ROSA) 26Sortm27.1(CAG-COP4*H134R/tdTomato) Hze/J strain (Stock No. 012567, Jackson Labs). The resulting offspring had the STOP cassette deleted in the heart, resulting in cardiomyocyte (MHC-ChR2) or Purkinje fibre (Cx40-ChR2) expression of the hChR2(H134R)-tdTomato fusion protein. MHC[Cre] mouse line was provided by Prof. Gillian Douglas (Radcliffe Department of Medicine, University of Oxford). Cx40[CreERT] was imported from Prof. Lucile Miquerol's Lab (Institut de Biologie du Développement de Marseille-Luminy).

### Tamoxifen treatment

Adult (week 8) mice (Cx40-ChR2 and *Dbh*[CreERT]/Rosa26-tdTomato transgenic mouse model) were injected with Tamoxifen (Sigma) (150 mg/kg, gavage) for 5 consecutive days. After one-week post-induction recovery, mice underwent subsequent experiment.

### Experimental design

Each of the developmental embryos or heart tissues were collected at specific time points, i.e. 8.5, 10.5, 12.5, 14.5 and P3. Samples are biological replicates to compared gene expression detected by SrT and scRNA-Seq. Consecutive tissue sections from the same heart tissue were considered technical replicates in the SrT (Fig. 1a, Fig. 2a) and RNAscope and immunohistology (Fig. 3c; Figs. 4–6, Figs. S1, S12) experiments. However, it is important to notice that consecutive sections are highly similar but not identical.

### Embryo and heart dissection

The whole embryos or hearts were dissected from developmental stages at E10.5, E11.5, E12.5, E13.5, E14.5, E16.5 and neonatal stage. Embryos or hearts were dissected in cold PBS (Life Technologies, CAT# 14190250), de-yolked and placed in PBS on ice until dissociation. The tissues were fixed with cold 4% paraformaldehyde (PFA) in 1× PBS for 15-20 minutes. After short fixation, the embryos and heart regions were either stored in 70% ethanol at 4 °C for future use or embedded in optimal cutting temperature (OCT) and frozen in cold isopentane cooled by dry ice and stored at −80 °C. Coronal cryosections (10 μm) were collected on a cryostat (Leica), mounted onto superfrozen glass slides (VWR) and stored at −80 °C. For adult heart sample preparation, the mice were killed by cervical dislocation, the heart was dissected and perfused PBS solution by through aorta. The heart was then fixed with cold 4% paraformaldehyde (PFA) in 1× PBS for no more than 30 min and moved into the 10% sucrose for 1 h, subsequently transferred to 20% sucrose for 1 h, and then cryoprotected into 30% sucrose for overnight at 4 °C. The heart was then embedded in OCT and frozen on isopentane-dry ice slurry and stored at −80 °C. Coronal cryosections (10 μm) were collected on a cryostat (Leica), mounted onto super-frost glass slides (VWR) and stored at −80 °C.

### 10× Genomics Chromium Single-cell RNA sequencing

The single cells were either isolated from embryos at E8.5 and E10.5 stages, or embryonic hearts at E12.5, E14.5 and E16.5 stages, and neonatal mouse cardiomyocytes (from P3) using the standard enzymatic method described previously[58]. Single-cell suspension samples were quality tested, quality control standards: cell activity > 80%, cell concentration 700-1200 cells/μL, cell diameter 5-40 μm, the total number of cells up to 500,000. Single cells were prepared following

the protocol from 10× Genomics, Inc (Pleasanton, CA). The protoplast suspension was loaded into Chromium microfluidic chips with 30 (v3) chemistry and barcoded with a 10× Chromium Controller (10× Genomics). RNA from the barcoded cells was subsequently reverse-transcribed and sequencing libraries constructed with reagents from a Chromium Single Cell 30 v3 reagent kit (10× Genomics) according to the manufacturer's instructions. Sequencing was performed with Illumina HiSeq according to the manufacturer's instructions (Illumina). FastQC was used to perform basic statistics on the quality of the raw reads.

Raw reads were demultiplexed and mapped to the reference genome by 10× Genomics Cell Ranger pipeline(https://support.10xgenomics.com/single-cell-geneexpression/) using default parameters. All downstream single-cell analyses were performed using Cell Ranger and Seurat[59] unless mentioned specifically. In brief, for each gene and each cell barcode (filtered by CellRanger[59]), unique molecule identifiers were counted to construct digital expression matrices in detail: cellranger count takes FASTQ files performs alignment, filtering, barcode counting, and UMI counting. It uses the Chromium cellular barcodes to generate feature barcode matrices.

To ensure robust and reliable transcriptomic signal-to-noise ratios, without impairing sensitivity to small signals, we filtered out all cells with unique RNA counts (nUMI) < 300, or distinct genes (nFeatures) < 270 to remove under-sampled cells and simple cells such as erythrocytes. The ratio of mitochondrial transcripts to nuclear genome-derived transcripts is often used as a metric of cell stress or quality in single-cell RNA sequencing. 5% is typically used as a ceiling, but this is not supported across cell types and can fail to identify damaged cells, particularly cardiomyocytes, which can reach ~30%[24] and exclude particular cardiomyocyte populations[60]. Given the changing nature of mitochondrial biogenesis across the embryonic to postnatal mouse heart[61], we utilized a dynamically changing filter for mitochondrial transcript ratios: E8.5: 5%, E10.5: 5%, E12.5: 7.5%, E14.5: 10%, E16.5: 15%, P3: 20%. We normalized cell libraries through SCTransform, which accounts for preservation of differential variation between highly variant and lowly variant genes[62]. We utilized Uniform Manifold Approximation and Projection (UMAP), following principal component analysis (PCA) and PCA dimension selection, to enable human-interpretable visualization of the transcriptomic space through dimensionality reduction. We visually examined the data for differences between batches of cells collected at each stage. Only P3 had significant batch differences. There are extensive suggested solutions for "correcting" batch differences[63]. However, from a statistical fundamentals perspective both a priori and empirically, such methods have been shown to produce aberrant downstream results[64] and so were not used here. We visualized UMAP in 3D and used Louvain clustering, and labelled clusters based on expression profiles. Further sub-clustering was performed as necessary, and a small number of cells were manually assigned where appropriate. Above was done using Seurat[65] functions unless specified.

To improve the discrimination of cardiomyocyte-lineage specific cell types in the Transcriptomic space, we isolated the cardiomyocyte-lineage cell types identified transcriptomically: Cardiac Progenitors, Early Cardiomyocytes, Immature Atrial Cardiomyocytes, Immature Ventricular Cardiomyocytes, Cardiac Conduction System, ECM Atrial Cardiomyocytes, ECM Ventricular Cardiomyocytes, Atrial Cardiomyocytes, Ventricular Cardiomyocytes, and Myh6[+] Ventricular Cardiomyocytes. We then re-normalised their raw nUMI counts, using SCTransform, and used PHATE[23] for dimensionality reduction and clustering. PHATE preserves global and local structure in the transcriptomic space of genes observed across cells: intuitive visualisation of high-dimensional gene expression data must balance within cell type (local) and between cell-type (global) transcriptional differences: local discrimination of transcriptional differences within cell types may be obscured by preservation of global between-cell type differences.

PHATE enables intuitive interpretation of datasets, particularly so for "lineage" type datasets in our experience. Cell types were assigned by comparison with literature expression profiles. We used the parallel implementation of Seurat's FindMarkers [https://github.com/vertesy/Seurat.multicore], implemented with min.pct=0.2, logfc.threshold=0.5, and test.use = "bimod" to compare differential gene expression between cell types. The 'bimod' option sets the use of a two-sided hypothesis test of the likelihood ratio model that models continuous and discrete aspects of gene expression between subjects[66]. $P$ values are then adjusted by a Bonferroni correction. In selecting $Dbh^+$ cells from this lineage, we isolated all cells from the cardiomyocyte lineage with a raw nUMI for $Dbh > 0$. We then re-normalised these cells, using SCTransform, and performed PHATE dimensionality reduction. We used cell labels identified from the cardiomyocyte-lineage. To identify whether $Dbh$ expression (defined as UMI > 0 or UMI = 0), was independent of cell type, we performed a two-tailed Chi Squared test for independence, using only cardiomyocyte lineage cell types composed primarily of cells beyond E10.5, as before this $Dbh$ expression was extremely limited in cell types. Therefore, we excluded Heart fields, Endocardial-gene rich cardiomyocytes, Developing cardiomyocytes, Misc., and Primary heart tube from testing.

## Stereo-seq spatial transcriptomics

The hearts at different stages (E12.5, E14.5, P3, and adult (P56)) were dissected from the wild type or $Dbh^{Cre}$/Rosa26-tdTomato mice and washed with PBS quickly. For adult heart dissection, the mice were killed by schedule 1, the heart was dissected and perfused PBS solution by through aorta and rings collected after embedded in Tissue-Teck OCT (Sakura, 4583). The heart samples were embedded in OCT. The OCT and frozen on isopentane-dry ice slurry and stored at −80 °C. Cryosectioning was performed with Leica CM1950 cryostat and the sections were cut with 10 µm thickness. Then the tissue sections were adhered to Stereo-seq chips immediately for the following experiments. Adjacent sections would be adhered to a glass slide to check tdTomato fluorescence.

Stereo-seq library preparation and sequencing were explained below as previously described[17,18]. Briefly, after tissue sections were adhered, Stereo-seq chips were incubated at 37 °C slide dryer for 3 min. Then the chips were immersed in prechilled methanol for 30 min. After fixation, tissue sections were stained with ssDNA reagent (Invitrogen, Q10212) for 5 min, and then washed using 0.1× SSC buffer (Ambion, AM9770) supplemented by 0.05 U/µl RNase inhibitor (0.1× SSC + RI). The ssDNA images were captured with Ti-7 Nikon Eclipse microscope using FITC channel. After imaging, tissue permeabilization was performed with 0.1% pepsin (Sigma, P7000) in 0.01 M HCl buffer (pH = 2) at 37 °C incubator for optimal time, which is 6 min for E12.5, E14.5 and P3 heart sections, and 18 min for adult heart sections. After washing with 0.1× SSC + RI, Reverse transcription was performed with SuperScript II (Invitrogen, 18064014, 10 U/µL reverse transcriptase, 1 mM dNTPs, 1 M betaine solution PCR reagent, 7.5 mM MgCl2, 5 mM DTT, 2 U/µL RNase inhibitor, 2.5 µM Stereo-TSO and 1× First-Strand buffer) at 42 °C incubator for overnight. After washing twice with 0.1× SSC buffer, tissues on the chip surface were digested with Tissue Removal Buffer (10 mM Tris-HCl, 25 mM EDTA, 100 mM NaCl, 0.5% SDS) at 37 °C incubator for 30 min. After washing with 0.1× SSC buffer for two times, the chips were treated with Exonuclease I (NEB, M0293L) at 37 °C incubator for 1 h. After washing once with 0.1x SSC buffer, cDNA was amplified using KAPA HiFi HotStart Ready Mix (Roche, KK2602) with 0.8 µM cDNA-PCR primer. For library preparation, cDNA products were further fragmented using in-house Tn5 transposase, amplified with KAPA HiFi HotStart Ready Mix, and purified using AMPure XP beads (Beckman Coulter, A63882). The purified PCR products were used to make DNBs (DNA nanoballs) which were sequenced with a MGI DNBSEQ-Tx sequencer.

Raw data processing method were explained below as previously described[23]. Briefly, CID and MID were extracted from the read 1, and the cDNA sequences were extracted from the read 2 in the FASTQ files from MGI DNBSEQ Tx sequencer. CID mapping allows 1 base mismatch. MID reads with N bases or more than two low quality score bases (score <10) were removed. The remaining DNA reads were aligned using STAR against the reference mouse genome (mm10) supplemented with WPRE sequences obtained from transgenic plasmid. To each gene of each CID, reads with same MIDs were only kept one for counting. After annotation and count, gene expression matrices with CID were generated.

Gene expression matrices were divided into bins with 20 × 20 DNBs for E12.5, E14,5, and P3 hearts, and 50 × 50 DNBs for adult heart. Density plot was used to visualize the gene count distribution in each bin, and the minor peak containing bins with low gene counts were excluded. Genes expressed in fewer than 5 bins were also discarded. After obtaining the final gene expression matrices of bin20 or bin50, data for all slices including three E12.5 slices, six E14.5 slices, and six P3 slices were merged to do the unsupervised clustering using Seurat. Briefly, SC Transform function was used to normalize the data and variable genes were identified. Then Run PCA was used for dimension reduction and run UMAP was used to indicate the two-dimensional projection with dims= 30. FindNeighbors and FindClusters were further used to identify clusters. Find All Markers was used to identify the differentially expressed genes for each group with the parameters: min.pct = 0.1, logfc.threshold = 0.25. Each cluster was annotated using several known cell type-specific marker genes. We assigned cluster labels using gene expression profiles and bin location.

To elucidate probabilistic spatial distributions of our cell types identified in our single -cell RNA sequencing data; we combined cells identified from the cardiomyocyte lineage with those identified in the wider dataset, to give a representative transcriptional reference library of potential cell types in the developing murine heart from E12.5 to P3. We combined: Atrial Cardiomyocytes, Ventricular Cardiomyocytes, Immature Atrial Cardiomyocytes, Immature Ventricular Cardiomyocytes, Immature Trabecular Ventricular Cardiomyocytes, Trabecular Ventricular Cardiomyocytes, Sinoatrial Node, Atrioventricular Node, Purkinje Fibres, Fibroblast-like, Smooth Muscle-like, Endothelium, Epicardium, Endocardium, Neural Crest, Haematopoetic Precursors, Immune Cells, and Platelets. We re-normalised the raw nUMI of this library using SCTransform[62]. To identify specific and robust marker genes for each cell type, we used SMASH[26], and in brief, kept the top 50 genes (Table S2) that contributed most to the classification of each cell type through identifying gene-wise Gini importance from the trained supervised ensemble machine learning classifier algorithm. We then used RCTD[67] to predict cell type composition of our spatial transcriptomics data through the use of a probabilistic classifier that decomposes spatial cell composition based on a single-cell RNA sequencing reference library.

## Immunofluorescence

The frozen sections were post fixed in 4% PFA fixation for 15 min at room temperature. The sections were washed in PBS for three times and 5 minutes for each wash, followed by permeabilization in 0.1% TritonX-100/PBS for 30 min. After three washes in PBS, the tissue sections were blocked by 0.1% Tween20, 10% Normal serum for 1 hour at room temperature. The sections were then incubated in the primary antibody for overnight at 4 °C. The slides were put in RT for 1 h for recovery. After three washes, the sections were incubated in the secondary antibody for 2 h. Finally, the mounting medium with DAPI were applied and nailed polish was used to seal the slides with coverslip. The antibodies used were as follows; anti-α-actinin (Abcam, Mouse, Ab9465, 1:200), anti-Th (Abcam, Rabbit, Ab6211, 1:200), Donkey anti-

Mouse 488 (Abcam, Ab150105, 1:200), Donkey anti-Rabbit 647 (Abcam, Ab150075, 1:200), Mounting-Media (Abcam, Ab104139, One droplet for each section), anti-CFP (Biovision, Rabbit, 5986-100, 1:100), anti-Flag (Novus Biologicals, Rat, NBP1-06712, 1:50).

## Multiplex nucleic acid in situ hybridization (RNAscope)

Each *RNAscope* experiment was replicated at least twice for identifying spatial expression of genes. The fixed frozen tissue slides were thawed at room temperature for 15 min and baked at 60 °C for 30 min. Afterwards, slides were post-fixed for 15 min in chilled 4% PFA, and serially dehydrated through 50%, 70%, 100%, and 100% ethanol for 5 minutes each. After dry for 5 minutes, the sections were applied the RNAscope hydrogen peroxide for 10 minutes. 1× Target retrieval reagent covered the sections for 5 minutes at 95 °C followed by two washes in distilled water. Tissue sections were permeabilized the protease digestion (ACD Protease III) for 20 minutes. The probes in C1 and C3 channels were labelled by using Opal 520, and 690 fluorophores (Akoya Biosciences, diluted 1:200) respectively. The C4 probe was developed using TSA biotin (TSA Plus Biotin Kit, Perkin Elmer, 1:500). Nuclei staining was performed with DAPI. Imaging was performed on a Zeiss Black software in the Zeiss confocal LSM780 microscope using 10× or 20× air-immersion objectives or 63 water-immersion objective. The excitation (Ex) laser and emission (Em) filter wavelengths were: DAPI (375 nm; 435-480 nm), Opal 520(488 nm; 500-550 nm), Opal 690 (676 nm; 690-700 nm). The probes used were as follow; Hcn4 (Mouse, Advanced Cell Diagnostics, Inc Cat No. 421271-C2), Dbh (Mouse, Advanced Cell Diagnostics, Inc Cat No. 407851-C3), Cacna2d2 (Mouse, Advanced Cell Diagnostics, Inc Cat No. 449221-C2), Id2 (Mouse, Advanced Cell Diagnostics, Inc Cat No. 445871-C2), Tbx18 (Mouse, Advanced Cell Diagnostics, Inc Cat No. 515221-C2), Shox2 (Mouse, Advanced Cell Diagnostics, Inc Cat No. 554291-C3), Th (Mouse, Advanced Cell Diagnostics, Inc Cat No. 317621-C4), Opal 520 (AKOYA BIOSCIENCES, FP1487001KT), Opal 690 (AKOYA BIOSCIENCES, FP1497001KT).Dilution for Probes were: C1:C2/C3/C4 is 1:50.

## 3D reconstruction of the heart

Three-dimensional image reconstruction was conducted to present the spatial architecture and morphology of cell distributions at the microscopic level (around μm) for the whole heart. The method was based on a modified procedure of our previous approach[68] to proceed fluorescence images of multiple coronal slices of adult and neonatal murine cardiac tissue. To establish a 3D atlas of whole heart with details of spatial distribution of positive and negative labelled cardiac cells, each two-dimensional (2D) fluorescence image was first pre-processed for registration, segmentation, and enhancement before final fusion for 3D reconstruction of the adult and neonatal hearts. Input: Raw 2D fluorescence images were used after spatial resolution between the two groups were unified (the resolution of the adult mouse cardiac images was downsized as they are about 20-fold of those of the neonate ones). Registration: Registration involves two steps: the contour extraction and the affine transformation. To account for the distinctively dispersed cell spatial distribution in the raw images, the Flood Fill algorithm was used to calculate numerous limited bounded regions, which corresponded to the position of holes. Once the positioning process was completed, a bitwise NOT inverted operation was performed to generate a new matrix, which was then logically computed with the raw image matrix to achieve the hole filling target. Finally, the Sobel operator method was utilized to construct a base-image emphasizing edge, followed which the new edge was then added to the former layer of each image to execute the affine transformation (rotation and translation) process. Contrast Enhancement: By modifying relative parameters gain (γ) and bias (β) in the following equation, we were able to optimize image features and boost colour contrast using a common linear transformation method: gx,y=γfx,y+β. In this equation, fx,y is the input pixel value, gx,y is the processed pixel value, the gain parameter γ is used to increase contrast ratio, and the bias parameter β is used to control brightness.

Once the multiple RGB images were enhanced, channel split measures were taken to separate them into red, green, and blue channels to implement the pixel aggregation operation. The specific procedures are bellowing: On the one hand, repeated the Hole Filling method (see the registration method) based on the blue channel and reassign all pixel values to 40 to create the basic contour. On the other hand, extracted red and blue channels and rewrite these pixel values separately to 180 and 10, and then clear the green channel values.

To construct the processed image and carry out a series of canonical images, we further merge these images with the same weight mechanism. Each image has a light grey base colour for the heart profile and a bright colour for a positive fluorescence cell. Frequency Analysis: With the purpose of quantify the pretreatment process, in this study we implemented the frequency analysis before 3D model building.

3D Reconstruction: Finally, all images were written to the same data file of VTK format (http://www.vtk.org/) to expedite hand visualizationbasedonParaviewsoftware (http://www.paraview.org/).

## Optogenetic Electrophysiology

Optical stimulation of ChR2 light-sensitive channel. Whole hearts were paced through the activation of ChR2 light-sensitive channels. This was achieved by the delivery of 470 nm blue light pulses (5–10 ms pulse width) generated by OptoFlash (Cairn Research). Pulses were triggered by a 1401 digitiser and Spike 2 software (Cambridge Electronic Design). Approximate blue light intensity was measured with a 818-ST2 Wand Detector connected to a 843 R Power metre (both Newport Corporation, CA, USA) and expressed normalised for the area being illuminated through simulating the average light intensity reached to the surface of the tissue by mimicking the distance of fibre-heart.

To monitor cardiac rhythms, we carried out ex vivo ECG analysis on langendorff perfused hearts. ECG parameters including RR interval, PR interval, QRS and QT durations were measured. Langendorff-perfused ex vivo hearts from *Dbh*^Cre^/ChR2 mice were subjected to programmed light stimulation (PLS) while ECGs were measured. Two protocols were carried out as follows: (a) continuous pacing protocol: Stimuli were delivered continuously with a constant frequency between 8 and 10 Hz. (b) S1S2 pacing protocol: a pacing train of eight stimuli (S1) was delivered at a basic cycle length of 100 ms, with a single (S2) premature extra stimulus introduced at progressively shorter intervals until the refractory period was reached.

Measurement of the Ventricular ERP. The ventricular ERP (VERP) of both the working cardiomyocytes and Purkinje fibres was evaluated in *Dbh*^Cre^-ChR2, MHC-Cre/ChR2-tdTomato and Cx40-CreERT/ChR2-tdTomato, respectively, by using S1S2 pacing protocol. The epicardial surface of the RV was photostimulated by a train of 10 light pulses (5–10 ms), at a frequency of 10 Hz (cycle length, 100 ms) (S1), followed by a premature optical stimulus (S2) at a progressively shorter coupling interval until the S2 failed to trigger ectopic beats at the VERP.

Pharmacological Purkinje Fibre Ablation. We carefully injected 500–1000 μl of Lugol's solution [5% (wt/v) iodine and 10% (wt/v) potassium iodide, in distilled water] into the RV cavity by using a Hamilton syringe supporting a 34 G needle followed by the continuous pacing protocol with RV epicardial light pacing.

## ECG recording and ventricular arrhythmias model in vivo

In vivo Surface ECG was recorded using a multichannel physiological instrument (MP150, BIOPAC Systems Inc, USA) under anaesthesia with 1–2% isoflurane using the gas anaesthesia machine (RWD Life Science, Shenzhen City, Guangdong Province, China).

## Echocardiography analysis in intact hearts

Mice were anesthetized with 1–2% isoflurane using a gas anaesthesia machine. Transthoracic M-mode echocardiographic recordings used the Vevo®3100 micro-ultrasound imaging system (FUJIFLIM VisualSonics Inc., Canada) following manufacturer's instructions. Three measurements taken at end-systole (s) and end-diastole (d) were averaged to calculate corresponding values of intraventricular septal thickness. Ejection fraction (EF) and fractional shortening (FS) were also acquired from the recorded measurements.

## Transmission electron microscopy (TEM)

The Dbh^Cre/tdTomato positive mouse (week 8) heart and adrenal tissue were harvested and examined under a fluorescent microscope to select the region showing tdTomato fluorescence. The tissues were fixed in 5% potassium dichromate/chromate pH 5.6 with the addition of formaldehyde to 4% for 48 h at 4 °C. Samples were then extensively washed in running deionized water overnight to remove any unstained solution, and stored at 4% formaldehyde before fully dehydrated through graded propan-2-ol (15 minutes each in 50%, 70%, 90%, 2 × 100%), infiltrated with LR White acrylic resin (30 minutes each in 50% LR White in propan-2-ol, 4 × LR White) and polymerized at 50 °C for 24 hours.

For light microscopy, 0.35 μm thick sections were stained with 0.1% toluidine blue in borax buffer. Sections were examined with an Olympus BX51 research light microscope (Olympus Optical Co Ltd., London, U.K.) and images captured with a Zeiss Axiocam digital camera and Axiovision software (Carl Zeiss Vision GmbH, Hallbergmoos, Germany).

For TEM examination, 100 nm thick sections were collected onto 150 mesh formvar/carbon-coated copper grids, stained with 2% uranyl acetate, and coated with a few nanometre layer of evaporated carbon to eliminate charging. For metal particle morphology, the samples were observed by a JEOL JEM2000 TEM (JEOL, Welwyn Garden City, UK), with STEM and EDAX Genesis 4000 Energy Dispersive X-Ray Spectrometry (EDAX, Leicester, UK) using 100 kV. Images were collected using a Getan Orius (11 megapixels) in-line digital camera (Gantan Inc. Abingdon, UK).

## Reporting summary

Further information on research design is available in the Nature Portfolio Reporting Summary linked to this article.

## Data availability

The *Dbh*^Cre, *Dbh*^CreERT, *Dbh*^CFP, *Dbh*^f/f mouse lines are available from M.L. and X.T. groups under a material transfer agreement with the University of Oxford and Southwest Medical University. Interactive versions of certain data are included with paper. Raw sequencing data generated by Stereo-seq and nucleic acid dye staining images have been deposited to Spatial Transcript Omics DataBase (STOmics DB) with the accession number STT0000009. The raw data of single cell RNA sequence have been deposited into CNGB Sequence Archive of China National GeneBank DataBase with accession number CNP0002316.

## Code availability

Representative code is available on github (https://github.com/Lei-group/Dbh_paper).

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

## Acknowledgements

We thank Peter Somogyi, FRS, FMedSci, PhD, FRS (Department of Pharmacology, University of Oxford) for help with imaging experiments and Wenyuan Zhang (Department of Zoology, University of Oxford) for help in preliminary bioinformatic analysis. We thank Prof. Gillian Douglas (Radcliffe Department of Medicine, University of Oxford) for providing MHC-Cre mouse line. We thank Niloufer G Irani and colleagues in Micron Advanced Bioimaging Unit at Department of Biochemistry (University of Oxford) for providing support and assistance for imaging work. We thank Prof. Christopher L-H Huang (Department of Physiology, University of Cambridge) for critical comments. We thank production team of China National GeneBank at Shenzhen for their assistance in providing Genebank resources. We thank Advanced Research Computing (ARC) service at University of Oxford for providing High Performance Computing (HPC) resource for the data process and analysis. We thank Dr. Tania Zaglia for scientific advice (Department of Biomedical Sciences, University of Padova). We thank Prof. V.M. Christoffels for providing advice on embryonic heart anatomy. This work was supported by the British Heart Foundation, UK [PG/23/11479] (M.L.) PG/22/11217 (M.L); PG/21/10512 (M.L.); FS/PhD/20/29053 (M.L., H.Z.); BHF Oxford Centre for Excellence (RE/18/3/34214), Medical Research Council, UK G100647 (M.L.), the National Natural Science Foundation of China [No. 81670310, X.T, No. 82270334, X.T, No. 81700308 X.O & No, 31871181, X.O, M.L.], the Science and Technology Department in Sichuan province of China [2020JDJQ0047 and 2022YFS0607 to X.T.], Wellcome Strategic Awards 091911/B/10/Z and (107457/Z/15/Z) (Micron Advanced Bioimaging Unit), NIHR01HL145060 (W.S.), NIHP01HL134599 (W.S.), NIHR01HL137036 (W.S.), and Additional Venture Research Fund (W.S., Y.L.).

## Author contributions

M.L. (lead contact) initiated and led the project, with input from W.S., X.T., designed the study, planned the experiments and supervised the study. M.L., W.S., T.Sun., A.G.R and X.T. interpreted the results and wrote and finalized the manuscript. T. Sun. conducted most immunohistochemistry, in situ hybridization experiments, imaging, and lineage analysis. A.G.R. analysed scRNA-Seq and Stereo-seq data, performed cell typing and prepared scRNA-Seq and Stereo-seq Figures. H.R., Z.P., K.K., X.O., T.C. and X.F. processed embryos and hearts for single-cell library preparation for scRNA-Seq. H.R., Z.P., T.C. processed embryos and hearts for Stereo-seq experiment. Y.A, X.G. jointly with A.G.R. prepared Stereo-seq data. S.T. conducted preliminary scRNA-Seq data analysis. Y.A, A.C, Z. S. provided support for Stereo-seq and data analysis. T.Sun., A.G.R M.L, Y.Li, Z.L., Z.P, Y.Z. conducted optogenetic electrophysiological experiments, data analysis and prepared the figure. Y.Liu., Y.Y.L. Performed embryonic tissue sectioning, immunohistochemical staining and the imaging. W.H., H.Z. performed three-dimensional image computational reconstruction. H.R., Z.P., T.Sun., Y.H., Y.R. and Y.Z. managed mouse lines. Z.X. performed imaging analysis. N.S. provided support for imaging and helped with manuscript preparation, L.M. provided Cx40cre and Cx40-GFP mouse lines and helped with data interpretation. All authors read and approved the final manuscript.

## Competing interests

The chip, procedure, and application of Stereo-seq are covered in pending patents. Y.A, A.C, Z. S. are employees of BGI and have stock holdings in BGI. The remaining authors declare no other competing interests.

## Additional information

**Peer review information** : *Nature Communications* thanks G.J. Boink and the other, anonymous, reviewer(s) for their contribution to the peer review of this work. A peer review file is available.

[1]Department of Pharmacology, University of Oxford, Mansfield Road, Oxford OX1 3QT, UK. [2]Key Laboratory of Medical Electrophysiology of the Ministry of Education, and Medical Electrophysiological Key Laboratory of Sichuan Province, Institute of Cardiovascular Research, Southwest Medical University, Luzhou, Sichuan 646000, China. [3]Department of Cardiology, the Affiliated Hospital of Southwest Medical University, Luzhou, Sichuan 646000, China. [4]Department of Physiology, School of Basic Medical Sciences, Southwest Medical University, Luzhou, Sichuan 646000, China. [5]BGI Research, Shenzhen 518103, China. [6]BGI Research, Qingdao 266555, China. [7]Department of Physics & Astronomy, The University of Manchester, Brunswick Street, Manchester M13 9PL, UK. [8]Herman B Wells Center for Pediatric Research, Department of Pediatrics, Indiana University School of Medicine, Indianapolis, USA. [9]Centre for Nanohealth, Swansea University Medical School, Swansea University, Singleton Park, Swansea SA2 8PP, UK. [10]Aix Marseille University, CNRS Institut de Biologie du Développement de Marseille UMR 7288, 13288 Marseille, France. [11]Department of Physiology, Anatomy & Genetics, Sherrington Building, Oxford, University of, Oxford OX1 3PT, UK. [12]Beijing Academy of Artificial Intelligence, 100084 Beijing, China. [13]These authors contributed equally: Tianyi Sun, Alexander Grassam-Rowe, Zhaoli Pu. ✉e-mail: tanxiaoqiu@swmu.edu.cn; wshou@iu.edu; ming.lei@pharm.ox.ac.uk

