## [Peer Review File · Nature Communications]

Dbh+ catecholaminergic cardiomyocytes contribute to the structure and function of the cardiac conduction system in murine heartEditorial Note: This manuscript has been previously reviewed at another journal that is not operating a transparent peer review scheme. This document only contains reviewer comments and rebuttal letters for versions considered at *Nature Communications*.

REVIEWER COMMENTS

Reviewer #5 (Remarks to the Author):

The authors utilize a rather extensive array of emerging and traditional methodologies to characterize expression of the dopamine beta-hydroxylase (Dbh) gene during cardiac development. They suggest that expression is enriched in several components of the cardiac conduction system, and then provide some indirect functional evidence using channel rhodopsins to implicate cells expressing Dbh in conduction system physiology.

Despite the impressive nature of the single cell analytics, including both spatial and more traditional methodologies, the most compelling and convincing data come from the RNAscope and immunofluorescent studies of Dbh expression in the developing heart. As noted in prior reviews, the single cell methods seem to lack the necessary resolution to provide any substantive additional information. For example, the plot in Figure 2c.ii doesn't show much in terms of resolving the spatial expression of Dbh and its putative enrichment in the CCS. I don't see how you can claim that it shows spatial co-localization of Dbh with the other ventricular CCS markers such as *Cntn2*, *Cacna2d2*, etc. While impressive, they don't move the biology forward to any meaningful degree. Moreover, enriched expression of Dbh has already been demonstrated in several published single cell and bulk RNA expression studies.

The channel rhodopsin studies are quite confusing to me. The authors state that stimulation of the MHC-driven ChR2 from the LV or the RV leads to a sinus like QRS. How can stimulation of an ectopic ventricular location produce a normal QRS morphology. If that is the case then it's hard to accept the other data provided with the Dbh-driven or Cx40-driven ChR2 constructs.

Given the existence of Dbh knockout mice and their cardiovascular phenotype, one wonders why the authors did not simply take advantage of their genetically engineered mice carrying fluorescent markers for Dbh expression and directly study their electrophysiology? It would have been quite useful to measure action potentials and other parameters in these cells and see how they compare to Purkinje cells or other conduction system cells.

The authors state that "In general, it has been widely accepted that the Purkinje fiber (PKJ) network arises from developing trabecular cardiomyocytes through endothelial-derived inductive signals. That may be true in avian development and reflect the important studies of Gourdie et al, where endothelial derived molecules such as endothelia were studied (PMID: 9618495), but this paper focuses on murine models, in which it is endocardially derived neuregulin signals that are thought be critical (PMID: 12149465).

Perhaps in response to earlier reviews, the authors go a bit overboard in terms of prior publications and should tone things down. For example, they mention several times "the superlative and pioneering works, such as that of Goodyer et al...."

Reviewer #6 (Remarks to the Author):

Sun et al. conducted an extensive study on a sub-population of Dbh+ cardiomyocytes, yet in my view failed to convincingly address the concerns of reviewer 2. The main issue with this work is that it describes a population of cells that were already identifiable based on publically available data-sets (in particular the work of Goodyer et al.). The presented collection of experiments largely illustrate the anatomical organization of this subpopulation and its association with the CCS (which also was already know based on the Goodyer data set). The lineage tracing experiments do not add much more information as they do not mark a population of progenitor cells, rather they mark spatial changes in expression that are not necessarily specific for Dbh (many other genes can show similar patterns), nor do they convincingly indicate that this is a specific cell type with a unique origin. The optogenetic experiment does not provide for much more mechanistic insight, it only shows that Dbh+ cardiomyocytes represent a subpopulation of cardiomyocytes that is present in close proximity or largely overlapping with CCS cells (which was predictable based on publically available data). That these cells subsequently show a similar optogenetic response as

compared to the Cx40-Chr2 is interesting in itself (although to be expected), yet does not provide much more functional insight into the role of Dbh+ cardiomyocytes. In this respect it would have been more informative to conduct a conditional knock-out of Dbh+ cardiomyocytes in adult mice followed by rigorous pharmacological and electrophysiological studies. Mechanistic single cell patch-clamp testing of Dbh+ working cardiomyocytes and/or Purkinje cells would also be highly informative by providing for more physiologic insight into the role of these sub-populations of cells.

Minor comments:

Line 374: Figure 6a \diamond Figure 5a

Figure 8: Lower panel lacks description in the legend

Figure 6D: MHC-Chr2: the typical tracings for the LV and RV appear not to indicate ventricular opto-pacing given the similar QRS morphology (which is not in line with the aggregate data); Dbh-Chr2: RA stimulation is not clear, this would benefit from additional local electrogram recordings to proof the atrial response to optopacing; the current tracing suggest sub-threshold stimulation as the ventricular response does not appear to correlate with the rate of optical stimulation; all ventricular opto-pacing interventions would benefit from a vector cardiogram analysis to further confirm correspondence with the site of ventricular stimulation.

REVIEWER COMMENTS

Reviewer #5 (Remarks to the Author):

Comment 1

The authors utilize a rather extensive array of emerging and traditional methodologies to characterize expression of the dopamine beta-hydroxylase (*Dbh*) gene during cardiac development. They suggest that expression is enriched in several components of the cardiac conduction system, and then provide some indirect functional evidence using channel rhodopsins to implicate cells expressing *Dbh* in conduction system physiology.

Despite the impressive nature of the single cell analytics, including both spatial and more traditional methodologies, the most compelling and convincing data come from the RNAscope and immunofluorescent studies of *Dbh* expression in the developing heart. As noted in prior reviews, the single cell methods seem to lack the necessary resolution to provide any substantive additional information. For example, the plot in Figure 2c.ii doesn't show much in terms of resolving the spatial expression of *Dbh* and its putative enrichment in the CCS. I don't see how you can claim that it shows spatial co-localization of *Dbh* with the other ventricular CCS markers such as *Cntn2*, *Cacna2d2*, etc. While impressive, they don't move the biology forward to any meaningful degree. Moreover, enriched expression of *Dbh* has already been demonstrated in several published single cell and bulk RNA expression studies.

Response:

We thank the Reviewer for their encouraging comments. With the retrospective view resulting from our functional and RNAscope/immunofluorescent investigations, we can see how the Reviewer might not deem our original scRNAseq/Stereo-Seq analysis to be adding much beyond them. However, we were sequentially investigating a signal that we first identified using these sequencing techniques, and thus presented our results starting from the beginning. Each technique answers a different question and helps us efficiently use resources.

We sought to use scRNAseq and Stereo-Seq in their most useful sense as hypothesis-generating techniques. We originally sought to investigate catecholaminergic cells in the developing mouse heart using scRNAseq – with the knowledge that this could provide a clue that would need further validation. We identified *Dbh* expression in the cellular context of the CCS. However, cardiac function requires not only cellular function but their appropriate regional localization: thus we used Stereo-Seq to localise our scRNAseq-identified cells within their spatial context. The use of both scRNA seq and Stereo-Seq enables us to balance the advantages and limitations of each technique, whilst minimizing over-reliance on any given technology.

We accept that the spatial plots of those gene maps do not provide a 1:1 co-localization of *Dbh* with the other CCS markers. Sadly, this is a current limitation of all spatially-resolved sequencing techniques that there is significant 'drop-out' of many genes, especially those with low abundance. However, Figure 2c.ii gave a global view of CCS-related gene profiles and their general regional co-localization with *Dbh*. For example, whilst *Dbh* is expressed across the heart, there are pockets of concentrated expression observed in a similar endocardial

region as pockets of *Id2* and *Cntn2* expression. If we had so desired, we could have used imputation techniques to recover a more aesthetically pleasing distribution. However, we are keen to show our data 'flaws and all'. Yet these techniques provided the original clue for us to investigate further in vitro, ex vivo, and in vivo. Thus, we then used RNAscope and lineage tracing (Figure 3-5) techniques to validate and complement Figure 2c.ii and demonstrate the colocalization between *Dbh* and *Cacna2d2* (Figure 3d) and CX40-GFP (Figure 5c) at a more granular level.

In our first reply to Reviewer 6 we more thoroughly cover the topic of whether signal measurement by other groups counts as identification. However, the bulk RNA sequencing does not have much relevance to the question we are investigating. Bulk digestion and sequencing of the heart/or regional myocardium that contains multiple cell types including sympathetic terminals and intrinsic cardiac adrenergic cells, both cell types abundantly express *Dbh* gene, would provide completely insufficient data to identify a novel cell type, and perhaps the view that it would be informative underlies why this biology has been unrecognized for so long.

Comment:

2

The channel rhodopsin studies are quite confusing to me. The authors state that stimulation of the MHC-driven ChR2 from the LV or the RV leads to a sinus like QRS. How can stimulation of an ectopic ventricular location produce a normal QRS morphology. If that is the case then it's hard to accept the other data provided with the *Dbh*-driven or *Cx40*-driven ChR2 constructs.

Response:

We thank the Reviewer for this helpful comment. We apologise for the ambiguous wording and presentation and have done additional analysis and revised the figure thoroughly and the main text to better distinguish between the regional and sub-regional pacing that we performed.

During our 2nd round of revision in 2022, according to the suggestions from Reviewer 4 (an expert in optogenetics and electrophysiology) as well as Reviewer 1, we conducted a series of new optogenetic electrophysiological experiments. These include:

1) introduced additional transgenic lines, *MHC-Cre/ChR2* (to label all cardiomyocytes) and *Cx40-Cre/ChR2* lines (to label the CCS), to compare with the electrophysiological properties of *Dbh-Cre/ChR2* (to label *Dbh*-derived cardiomyocytes).

2) applied focal epicardial photostimulation to determine cell type dependent responses of *Dbh-Cre/ChR2* cells as used by Zaglia *et al* (*PNAS*, 2015 112 (32) E4495-E4504). In their paper, the authors applied focal epicardial photostimulation to determine cell type dependent responses. Zaglia *et al* observed that uniform response to photoactivation of the *MHC-Cre/ChR2* ventricles. Whereas, Purkinje fiber stimulation using the *Cx40-Cre/ChR2* line yielded variable electrical response depending on the illumination site. Ectopies triggered by photoactivation of the atrioventricular junctional CCS, had QRS durations indistinguishable from sinus rhythm QRS complexes, which is consistent with the expected ECG morphologies from the different conduction system regions. We adapted the methods published by Zaglia *et al* (*PNAS*, 2015 112 (32) E4495-E4504) to determine Purkinje fiber function *in vivo* using the *Cx40-Cre/ChR2* line and comparing with both *MHC-Cre/ChR2* and *Dbh-Cre/ChR2* lines. Thus, we adopted their protocols and conducted focal epicardial photostimulation to determine cell type-dependent response of *Dbh-Cre/ChR2* cells in comparison with *Cx40-Cre/ChR2* and *MHC-Cre/ChR2* lines. We observed that *Dbh-Cre/ChR2* hearts showed similar effects

responding to the focal epicardial photostimulation as Cx40-Cre/ChR2 hearts, but different to MHC-Cre/ChR2 hearts. The results are consistent with Zaglia *et al*/ reported in MHC-Cre/ChR2 and the Cx40-Cre/ChR2 lines

We agree that the wording used to describe this overall pacing response and QRS waveforms from MHC-Cre/ChR2 hearts (the control line we used) causes confusion and is misleading. What we meant was that 1) MHC-Cre/ChR2 hearts have a response from all four chambers and all 6 subregions assayed in the heart 2) uniform QRS morphologies in response to photoactivation in the MHC-ChR2 ventricles, whereas these are more variable in than those regions in *Dbh*-ChR2 and Cx40-ChR2 hearts. We have now revised and expanded our Figure 6 to more explicitly demonstrate the above. We now include sinus rhythm traces at the top of the figure, and attempt to include a sinus rhythm activation in each trace prior to light pacing where possible, and include the labelling of the difference between the subregional light pacing we performed on ventricular bases and apices. We again, thank the Reviewer for highlighting this confusion and believe the new Figure 6 explains our point much more clearly.

We revised the original sentences between 420-425 to:

“We compared the regional responsiveness and ECG waveform morphologies of Langendorff-perfused hearts from three lines measured at sinus rhythm or light pacing at cycle length 100-120 ms. MHC-Cre/ChR2-tdTomato (MHC-ChR2) hearts show uniform responsiveness to photostimulation with blue light in all four chambers (LA, RA, LV, RV) of the heart, and all 6 subregions of the heart, with each ventricle separated into its base and apex (Figure 6a). However, *Dbh* (*Dbh*-Cre/ChR2) and Cx40 (Cx40-Cre/ChR2) hearts predominately respond to blue light pacing of the atria and the basal RV (Figure 6a). However, *Dbh* (*Dbh*-ChR2) and Cx40 (Cx40-ChR2) hearts often show wider and variable QRS waveforms than that of MHC-ChR2 hearts in specific region of the heart, specifically the right ventricle apex

We also now change the Figure legend to read:

“Figure 6. Direct optogenetic assessment of Purkinje fiber function implicates *Dbh*⁺-derived CMs in the ventricular CCS. a) Representative ECG traces from Langendorff-perfused hearts subjected to localized light pacing in four different chambers and 6 different subregions of hearts (SR: sinus rhythm, LA: left atrium, RA: right atrium, RVbase: right ventricle base, LVbase: left ventricle base, RVapex: right ventricle apex, LVapex: left ventricle apex) with blue light isolated from optogenetic transgenic mouse lines expressing the fused protein tdTomato-ChR2 under the control of the promoters for MHC (MHC-ChR2), *Dbh* (*Dbh*-ChR2) and Cx40 (Cx40-ChR2). Blue triangles indicate when blue light pacing was applied. b) A diagram displaying the electrical responsiveness of the different heart chambers to similar regional optical pacing across the different mouse genetic lines. c) Comparison of effective refractory periods (ERP) determined by RV epicardial optical programmed stimulation in hearts from three optogenetic transgenic mouse lines. (n=5-8 per group) * p<0.05, ** p< 0.01, + p>0.05. d) Representative ECG traces of ectopic beats originated by epicardial light stimulation of the RV from three optogenetic transgenic mouse lines before and after (5 min) RV intracavitary Lugol’s solution injection (n = 5-6 mice per mouse line). Blue arrows indicate the light pulses. Lugol’s solution treatment caused enlargement of the QRS complex and abolished light-induced ectopies in *Dbh*-ChR2 and Cx40-ChR2, but not MHC-ChR2 hearts.”

Comment

Given the existence of *Dbh* knockout mice and their cardiovascular phenotype, one wonders

3

why the authors did not simply take advantage of their genetically engineered mice carrying fluorescent markers for *Dbh* expression and directly study their electrophysiology?

Comment 4

It would have been quite useful to measure action potentials and other parameters in these cells and see how they compare to Purkinje cells or other conduction system cells.

Response:

We have combined these two comments in the name of alacrity. However, we believe that the first point relates to why we didn't use previously existing *Dbh* KO mice.

- 1) The existing *Dbh* knockout mouse model is a global gene knockout. Although the model presented some interesting cardiac phenotypes, it is not possible to use this model to determine the function of *Dbh*⁺ cardiomyocytes specifically.
- 2) Our functional studies sought to understand the role of *Dbh*⁺ cardiomyocytes in the context of their potential regulatory role in cardiac function. Isolated studies of cardiomyocytes' electrophysiology are not expected to reveal differences between *Dbh*⁺ cardiomyocytes and their neighbouring *Dbh*⁻ cardiomyocytes. Furthermore, any difference might be related to autocrine/paracrine action and thus be hard to control for adequately. However, as we expected the major difference to be related to their potential role in intracardiac intercellular communication, we sought to use intact cardiac preparations to investigate any functional differences.
- 3) Following previous Reviewer 2's suggestion, we generated a mouse model with cardiomyocyte specific deletion of *Dbh* gene by using cardiac-specific alpha myosin-heavy chain (*Myh6*) promoter to drive expression of Cre. Coincidentally, we got this model recently: *Dbh*^{CKO} mice show comparable ECG and echocardiographic characteristics to their control littermates (*Dbh*^{fl/fl} mice), indicating that cardiomyocyte-specific deletion of *Dbh* does not affect cardiac electrical and mechanical function at baseline condition in vivo. However, isolated *Dbh*^{CKO} hearts as Langendorff-perfused preparations, and thus without autonomic regulation, have significant longer: P-R intervals, AVN effective refractive periods (AVNERPs) and atrial-ventricular (A-V) conduction Wenckebach block periods. These results suggest that cardiomyocyte *Dbh*-expression contributes to baseline intracardiac conduction regulation, but that at baseline the autonomic nervous system can account for its deficit. Any isolated study of cardiomyocytes and their electrophysiological and/or secretory function, would have been informative to a lesser degree and not indicative of whether these cells have any importance in baseline function at the level of the whole organ.
- 4) Nonetheless, we appreciate the Reviewer's excellent suggestion. Given the complexity of the CCS: the highly heterogeneous cell types such as AVN, His bundle, and Purkinje fibers, makes isolating these cells regionally in a cell-type specific manner a huge task. We are actually one of the first groups to isolate mouse SAN cells (Lei et al *J Physiol.* 2004;559: 835-48), it took us more than three years to complete the work for characterization mouse SAN cells. Isolation of AVN or His bundle CCS cells from mouse heart would require extensive methodological development. To our best knowledge, we have not seen any report about systemic characterization of the electrophysiology of AVN, His bundle, Purkinje fibers in mouse heart. Therefore, while we appreciate the Reviewer's excellent suggestion, this remains to be addressed in the future.

Comment 5

The authors state that “In general, it has been widely accepted that the Purkinje fiber (PKJ) network arises from developing trabecular cardiomyocytes through endothelial-derived inductive signals. That may be true in avian development and reflect the important studies of Gourdie et al, where endothelial derived molecules such as endothelia were studied (PMID: 9618495), but this paper focuses on murine models, in which it is endocardially derived neuregulin signals that are thought be critical (PMID: 12149465).

Response: We thank the Reviewer for their excellent suggestion. We have revised the relevant text as:

“In murine hearts, it has been accepted that the Purkinje fiber (PKJ) network arises from developing trabecular cardiomyocytes through endocardially-derived inductive signals.”

Comment 6

Perhaps in response to earlier reviews, the authors go a bit overboard in terms of prior publications and should tone things down. For example, they mention several times "the superlative and pioneering works, such as that of Goodyer et al...."

Response: We thank the Reviewer for their excellent suggestion, we have revised the text and reduced repeated citations.

Reviewer #6 (Remarks to the Author):

Comment 1

Sun et al. conducted an extensive study on a sub-population of Dbh+ cardiomyocytes, yet in my view failed to convincingly address the concerns of reviewer 2. The main issue with this work is that it describes a population of cells that were already identifiable based on publically available data-sets (in particular the work of Goodyer et al.). The presented collection of experiments largely illustrate the anatomical organization of this subpopulation and its association with the CCS (which also was already know based on the Goodyer data set). The lineage tracing experiments do not add much more information as they do not mark a population of progenitor cells, rather they mark spatial changes in expression that are not necessarily specific for Dbh (many other genes can show similar patterns), nor do they convincingly indicate that this is a specific cell type with a unique origin.

Response:

We thank the Reviewer for their comments. However, as we responded to the previous Reviewer #2, We don't agree that Goodyer et al identified this cell type. Goodyer et al make no attempt to identify this cell type at all, for example, no Figure about expression of Dbh gene in the Results section and discussion on the gene. Goodyer et al only present the measurement of a signal in their supplementary data, concurrently with over 2200 other genes, and make no reference, highlight, or any other attempt to in any way discuss or differentiate *Dbh* from any background signal in their entire sequencing assay (which obviously covers the majority of the known gene universe through scRNAseq). Goodyer et al, and others, simply record the signal of this gene included in the rest of their background signals. We would

suggest that the work of, and similar to, Goodyer et al indeed provided some hope to us personally during the early stages that it was not likely just a sequencing read misattribution during our scRNAseq, but that remains the sum of the involvement of that work with this specific discovery: we were the first to identify the potential of this near-background signal and identify that it was indeed expressed within cardiomyocytes and not merely free-floating contaminant mRNA, or cell-wise contamination in the form of doublets, such as sympathetic neuron contamination with a cardiomyocyte. We believe that recording a signal amongst tens of thousand other signals, and making no attempt to differentiate or note it, to not constitute identification but merely a recording that is undifferentiated from noise. We highlight the work of these other groups in our manuscript. However, we believe that both philosophically and practically we are the first group to identify this signal as being of interest, and to then describe this signal and use it as a basis for further experimentation and interrogation that results in the discovery of new biology.

Comment 2

The optogenetic experiment does not provide for much more mechanistic insight, it only shows that Dbh+ cardiomyocytes represent a subpopulation of cardiomyocytes that is present in close proximity or largely overlapping with CCS cells (which was predictable based on publicly available data). That these cells subsequently show a similar optogenetic response as compared to the Cx40-Chr2 is interesting in itself (although to be expected), yet does not provide much more functional insight into the role of Dbh+ cardiomyocytes. In this respect it would have been more informative to conduct a conditional knock-out of Dbh+ cardiomyocytes in adult mice followed by rigorous pharmacological and electrophysiological studies. Mechanistic single cell patch-clamp testing of Dbh+ working cardiomyocytes and/or Purkinje cells would also be highly informative by providing for more physiologic insight into the role of these sub-populations of cells.

Response: We thank the Reviewer's astute suggestion for a cardiomyocyte-specific conditional knockout of *Dbh*. We planned this model for follow up study on *Dbh*⁺ cardiomyocytes and successfully established the model recently: we generated a mouse model with cardiomyocyte specific deletion of *Dbh* gene we generated a mouse model with cardiomyocyte specific deletion of *Dbh* gene by using cardiac-specific alpha myosin-heavy chain (*Myh6*) promoter to drive expression of Cre. We have conducted both in vivo and ex vivo electrophysiological characterization. The new results are now presented in the new main Figure 7 and Supplementary Figures S20, S22. In sum, *Dbh*^{CKO} mice show comparable ECG and echocardiographic characteristics to their control littermates (*Dbh*^{fl/fl} mice), indicating that cardiomyocyte-specific deletion of *Dbh* does not affect cardiac electrical and mechanical function at baseline conditions in vivo. However, isolated *Dbh*^{CKO} hearts as Langendorff-perfused preparations, and thus without autonomic regulation, have significantly longer: P-R intervals, AVN effective refractive periods (AVNERPs) and atrial-ventricular (A-V) conduction Wenckebach block periods. These results suggest that cardiomyocyte *Dbh*-expression contributes to baseline intracardiac conduction regulation, but that at baseline the autonomic nervous system can account for its deficit.

2. We agree with glee that it would be interesting to do single-cell studies. However, this involves methodological expansion that will require multiple years of further study. Single-cell electrophysiological characterization, as we explained to Reviewer 1, mandates the development of new methodologies. Given the complexity of the CCS: the highly heterogeneous cell types such as AVN, His bundle, and Purkinje fibers, makes isolating these cells regionally in a cell-type specific manner a huge task and not feasible for this paper. We

are actually one of the first groups to isolate mouse SAN cells (Journal of Physiology 2004), and it took us more than three years to complete the work for characterization mouse SAN cells. Isolation of AVN or His bundle CCS cells from mouse heart would require extensive methodological development. To our best knowledge, we have not seen any report about systemic characterization of the electrophysiology of AVN, His bundle, Purkinje fibers in mouse heart. Therefore, while we appreciate the Reviewer's excellent suggestion, this remains to be addressed in the future.

Thus, it remains to be elucidated whether these cells are truly indistinguishable from other CCS cells electrophysiologically. However, we can already determine that their autocrine/paracrine secretion has an important functional role in intracardiac conduction at baseline. There are a number of pathological conditions where the prospective pertinence of this new paradigm of catecholaminergic CMs warrants further investigation in new light, such as CPVT, heart failure, atrial fibrillation, or post-orthotopic heart transplant. For example, it is interesting to speculate whether these *Dbh⁺*-CMs might contribute to atrial fibrillation pathology through increased likelihood of passing atrial fibrillatory waves through to the ventricles in a patient-specific manner.

Minor comments:

Line 374: Figure 6a \diamond Figure 5a

Response: We thank the Reviewer for spotting this error. It has now been corrected.

Figure 8: Lower panel lacks description in the legend

Response: We thank the Reviewer for spotting this error. More legend has now been added.

Figure 6D: MHC-ChR2: the typical tracings for the LV and RV appear not to indicate ventricular opto-pacing given the similar QRS morphology (which is not in line with the aggregate data); Dbh-ChR2: RA stimulation is not clear, this would benefit from additional local electrogram recordings to proof the atrial response to optopacing; the current tracing suggest sub-threshold stimulation as the ventricular response does not appear to correlate with the rate of optical stimulation; all ventricular opto-pacing interventions would benefit from a vector cardiogram analysis to further confirm correspondence with the site of ventricular stimulation.

Response: We thank the Reviewer for their suggestion.

We apologise for the confusion, the waterfall 'aggregate data' was not from all hearts but from a specific heart. With respect to the Dbh RA comment and sub-threshold stimulation, the trace for Dbh-ChR2 RA light pacing shows what is commonly referred to as 'T-on-P', where the P wave is slightly masked by the T wave. Due to space constraints we can only include two-to-three light-induced beats per trace, whereas full traces would much more markedly demonstrate the ventricular response to this higher pacing frequency from RA stimulation. However, you can note the loss of the sinus P wave during the pacing, being replaced by the T-on-P P waves compared to the beat included just before pacing on the same trace, and you can see that the two QRS complexes induced by pacing are closer together than the sinus QRS to the first light-induced QRS and that this duration between the pacing-induced QRS complexes is the same as the pacing interval.

We have now revised this figure in line with recommendations from Reviewer #1 as above. We have revised and expanded the figure to now better represent the true stimulations of the different chambers and subregions. Our previous representative traces included some examples of optical pacing-induced arrhythmia, which we originally intended to discuss further but did not in the main text. Thus, we have now replaced certain representative images and now include some examples of sinus rhythm traces from each heart, and traces from the subregional ventricular pacing we performed to examine the difference between apical and basal responses.

Unfortunately, vector cardiogram analysis requires a multi-lead electrocardiogram set-up which is not a common practice nor feasible in isolated small rodent (especially mouse) hearts. Whilst vector cardiogram analysis is useful in clinical practice and for in vivo multi-lead set-ups, isolated hearts are limited by 1) no consensus on reliable or reproducible electrode placement for lead 1 or 3 electrodes 2) physical space as electrodes can be large relative to the heart size and they would compete for space around small mouse hearts. A small number of niche and specialised multi-electrode arrays do exist. However, these number in the single digits and would preclude the even illumination of the heart with the blue pacing light. Thus, as is common practice for reliable and reproducible ECGs from murine hearts, we used a lead II electrode placement that measures the major vector from the RA through to the left ventricular apex of the heart. With only one lead it is impossible to separate the vector into its isolated horizontal or vertical components. Furthermore, to our knowledge, there is no work that has correlated any such vector cardiograms with ectopic origin in murine hearts, and thus this would require extensive methodological development and validation to establish what the murine correlates are for this in-human clinical technique of vector cardiogram analysis.

REVIEWERS' COMMENTS

Reviewer #5 (Remarks to the Author):

The authors have done a significant amount of additional work to address the concerns of the many reviewers. The additional mutant models and physiology provide some evidence that Dbh regulates cardiac electrophysiology, although the mechanistic basis for this effect will likely have to wait for additional studies. I still have some lingering concerns related to the implications of the ontogenetic studies. The explanation provided is helpful, but incomplete. On a minor note, the authors should update reference 5 or 6 with references relevant to murine CCS development, not avian CCS development, such as PMID:12149465 and should include the original references for some of the CCS markers such as Id2 (PMID17604724), Cntn2 (PMID20110552) and others that are described in this study.

Reviewer #6 (Remarks to the Author):

My comments have sufficiently been addressed.

Response to Reviewers' comments

Reviewer #5 (Remarks to the Author):

Comment:

The authors have done a significant amount of additional work to address the concerns of the many reviewers. The additional mutant models and physiology provide some evidence that *Dbh* regulates cardiac electrophysiology, although the mechanistic basis for this effect will likely have to wait for additional studies.

Response:

We are grateful for your accurate summary of the improvement of our paper and the constructive comments.

I still have some lingering concerns related to the implications of the ontogenetic studies. The explanation provided is helpful, but incomplete.

Response:

We appreciate your comment. We further improved the explanation of optogenetic studies in the results section to provide more explanations as the highlighted text below in page 13

Firstly, we determined if *Dbh*⁺-derived CMs are associated with ventricular CCS, we compared photostimulation-induced electrophysiological characteristics of *Dbh*^{Cre}/ChR2-tdTomato hearts with Cx40-CreERT/ChR2-tdTomato and MHC-Cre/ChR2-tdTomato hearts. Cx40-CreERT /ChR2-tdTomato mice have been used as an optogenetic tool for studying ventricular CCS by expressing ChR2 in Purkinje fibers²⁸, while MHC-Cre/ChR2-tdTomato line has been used for studying cardiomyocytes in general as a non-selective cardiomyocyte ChR2 expressing mouse model. Despite such technique provide specific focal epicardial photostimulation to determine cell type dependent responses, due to tight electronic connection between ChR2 expressed cells and non-ChR2 expressing cells, such cell-type dependent responses may not be accurate. To enable timely and spatially controlled photostimulation of ChR2 hearts, we used a fiber optic delivering 470 nm light pulses (5 ms), generated by a time-controlled light emitting diode (LED) directed towards the epicardium of Langendorff-perfused hearts in LA, RA, LV and RV regions (Fig.6a). ECGs were then recorded for electrophysiological analysis. We compared the regional responsiveness to photostimulation and ECG waveform morphologies of Langendorff-perfused hearts from three lines measured at sinus rhythm or light pacing at cycle length

100-120 ms. MHC^{Cre}/ChR2-tdTomato (MHC-ChR2) hearts show uniform responsiveness to photostimulation with blue light in all four chambers (LA, RA, LV, RV) of the heart, and all 6 subregions of the heart, with each ventricle separated into its base and apex (Figure 6a). However, Dbh^{Cre}/ChR2-tdTomato (Dbh-ChR2) and Cx40^{Cre}/ChR2-tdTomato (Cx40-ChR2) hearts predominately respond to photostimulation of the atria and the basal RV (Figure 6a). However, Dbh-ChR2)and Cx40-ChR2 hearts often show wider and variable QRS waveforms than that of MHC-ChR2 hearts. This demonstrates that Dbh-ChR2 hearts have a similar photostimulation response to Cx40-ChR2 hearts, in contrast to MHC-ChR2 hearts, in terms of chamber-specificity and RV light pacing (LP)-induced QRS waveform characteristics. In contrast to the uniform response to photoactivation of the α -MyHC-ChR2 ventricles, Purkinje fiber stimulation using the Cx40-Cre/ChR2 line yielded variable electrical response depending on the illumination site, particularly ectopies triggered by photoactivation of the atrio-ventricular bundle, had QRS duration identical to the spontaneous complex, which is consistent with the physiological characteristics of the different conduction system regions.

On a minor note, the authors should update reference 5 or 6 with references relevant to murine CCS development, not avian CCS development, such as PMID:12149465 and should include the original references for some of the CCS markers such as Id2 (PMID17604724), Cntn2 (PMID20110552) and others that are described in this study.

Response:

We appreciate your comment. We updated references 5 and 6 and added additional references as you suggested.

Reviewer #6 (Remarks to the Author):

My comments have sufficiently been addressed.

Response:

We are grateful for your recognition of our effort to address the issues raised in your review report and of the improvement of our paper.